# Carob pod polyphenols suppress the differentiation of adipocytes through posttranscriptional regulation of C/EBPβ

**Kasumi Fujita[1], Toshio Norikura[2], Isao Matsui-Yuasa[1], Shigenori Kumazawa[3], Sari Honda[3], Takumi Sonoda[4], Akiko Kojima-Yuasa**[1]*

**1** Department of Food and Human Health Sciences, Graduate School of Human Life Science, Osaka City University, Osaka, Japan, **2** Department of Nutrition, Aomori University of Health and Welfare, Aomori, Japan, **3** Department of Food and Nutritional Sciences, University of Shizuoka, Shizuoka, Japan, **4** TAISHO TECHNOS, Co., Ltd., Tokyo, Japan

* kojima@life.osaka-cu.ac.jp

**Data Availability Statement:** All relevant data are within the paper and its Supporting Information files.

## Abstract

Obesity is a major risk factor for various chronic diseases such as diabetes, cardiovascular disease, and cancer; hence, there is an urgent need for an effective strategy to prevent this disorder. Currently, the anti-obesity effects of food ingredients are drawing attention. Therefore, we focused on carob, which has high antioxidant capacity and various physiological effects, and examined its anti-obesity effect. Carob is cultivated in the Mediterranean region, and its roasted powder is used as a substitute for cocoa powder. We investigated the effect of carob pod polyphenols (CPPs) on suppressing increases in adipose tissue weight and adipocyte hypertrophy in high fat diet-induced obesity model mice, and the mechanism by which CPPs inhibit the differentiation of 3T3-L1 preadipocytes into adipocytes *in vitro*. In an *in vivo* experimental system, we revealed that CPPs significantly suppressed the increase in adipose tissue weight and adipocyte hypertrophy. Moreover, in an *in vitro* experimental system, CPPs acted at the early stage of differentiation of 3T3-L1 preadipocytes and suppressed cell proliferation because of differentiation induction. They also suppressed the expression of transcription factors involved in adipocyte differentiation, thereby reducing triacylglycerol synthesis ability and triglycerol (TG) accumulation. Notably, CPPs regulated CCAAT/enhancer binding protein (C/EBP)β, which is expressed at the early stage of differentiation, at the posttranscriptional level. These results demonstrate that CPPs suppress the differentiation of adipocytes through the posttranscriptional regulation of C/EBPβ and may serve as an effective anti-obesity compound.

## Introduction

Obesity is a major risk factor for hyperglycemia, high blood pressure, and arteriosclerosis, leading to various chronic diseases such as diabetes, cardiovascular disease, and cancer. Therefore, it is very important to prevent obesity. Currently, the anti-obesity effects of food ingredients are drawing attention. Obesity is a condition in which adipose tissue accumulates triacylglycerol (TG) because of an increase in the number of adipocytes and enlargement of

**Funding:** This work was supported by JSPS KAKENHI (Grant Number JP15K00832). KAKENHI aims to significantly develop all "academic research" from basic to applied (research based on the free thinking of researchers). TAISHO TECHNOS, Co., Ltd. provided the Carob pod polyphenols, a part of the grant, and support in the form of salary for T.S., but did not have any additional role in the study design data collection and analysis decision to publish, or preparation of the manuscript. The specific role of T.S. is articulated in the "Author Contributions" section.

**Competing interests:** A part of research grant and Carob pod polyphenols were provided by the TAISHO TECHNOS, Co., Ltd. We declare that these relationships did not affect the results and conclusions of this manuscript. This does not alter on adherence to PLOS ONE politics on sharing data and materials.

adipocytes. Adipose tissue is an endocrine organ that secretes various adipocytokines and serves as an energy storage organ [1]. It changes the balance of adipocytokine secretion depending on the size of adipocytes and contributes to the development of lifestyle-related diseases and atherosclerosis. Therefore, a reduction in the number of adipocytes and suppression of adipocyte hypertrophy are important anti-obesity effects.

Carob (*Ceratonia siliqua* L.) is an evergreen tree of the genus Locust and is cultivated mainly in the Mediterranean region. After roasting carob pods, the crushed remnant is called carob pod powder, which is used as a substitute for cocoa powder because of its color and flavor [2]. Carob pods also contain polyphenols and minerals, such as calcium, phosphorus, and potassium. Previously, it was reported that *in vivo* and *in vitro* antioxidant activity is associated with the polyphenol concentration in carob pod extracts [3]. Therefore, the antioxidant activities of their polyphenols are thought to be associated with the various physiological and pharmacological actions of Carob pods. In an *in vitro* study using a β-carotene bleaching assay, Kumazawa *et al.* reported that carob pod polyphenols (CPPs) had stronger antioxidant effects against the discoloration of β-carotene than catechins and procyanidins [3]. Furthermore, several *in vivo* models have demonstrated that carob pods have hepatoprotective effects against carbon tetrachloride-induced toxicity [4] or ethanol-induced liver injury [5], gastroprotective effects against ethanol-induced oxidative stress [6], and antidiabetic effects against alloxan-induced diabetes [7] or streptozotocin-nicotinamide-induced diabetes [8]. However, the anti-obesity effect and its mechanism of action have not been clarified.

In humans and rodents, it has been suggested that an increase in free and saturated fatty acids by consuming a high fat diet enhances adipogenesis and induces an inflammatory response underlying metabolic syndrome [9, 10]. Therefore, examining the anti-obesity effect of carob pod extract under the intake of a high fat diet in mice is very important to determine the effect of carob pod extract in preventing metabolic syndrome. In the present study, we used a polyphenol-rich fraction eluted with methanol from an HP-20 adsorption resin column as the CPPs.

CPPs have been reported to have preventive effects against liver injury and antidiabetic effects due to their strong antioxidant activity. However, it is not clear whether CPPs can suppress obesity, which causes lifestyle-related diseases. Therefore, the purpose of the present study was to clarify whether CPPs have an anti-obesity effect using an *in vivo* mouse model and an *in vitro* cell model. First, the presence or absence of the anti-obesity effect of CPPs was confirmed using an *in vivo* model with a high fat-induced model. The mechanism by which CPPs inhibit the differentiation of preadipocytes into adipocytes was investigated using 3T3-L1 preadipocytes.

## Materials and methods

### Preparation of CPPs

CPPs were produced by roasting and crushing carob pods in an oven at 120 to 180˚C. The carob powder with a roasting time of 30 min was called "Light", and the sample with a roasting time of 60 min was called "Extra Dark". Each powder (500 g) was extracted with 2.5 L of methanol at room temperature overnight. Suction filtration was then performed, and the residue was further extracted with methanol with stirring at room temperature overnight. Similarly, after suction filtration, the extracts were combined, and the solvent was removed using an evaporator. The yields of carob pod-Light and carob pod-Extra Dark were 188 and 107 g, respectively. Next, each extract was dissolved in a small amount of water-containing methanol and applied to an open column (5.5 × 50 cm) packed with conditioned HP-20 adsorption resin. The column was repeatedly passed with water to remove water-soluble components

such as sugars and amino acids. After that, methanol flowed over the column to elute the polyphenol component adsorbed on the resin. The methanol eluate was evaporated and dried on a vacuum pump. The yields of carbo pod-Light and carbo pod-Extra Dark were 13.7 and 19.1 g, respectively. Total polyphenol content was determined by the Folin-Ciocalteu method [11], and the total polyphenol concentrations for CPP (Light) and CPP (Extra Dark) were 30.9 ± 1.8% and 28.0 ± 1.5%, respectively.

## Animal treatment

The study was approved by the Ethics Committee of Laboratory Animals, and the analysis using laboratory animals complied with the regulations of the Osaka City University Laboratory Animal Committee (permission number: S0056). Thirty-six male C57BL/6J mice (8 weeks old) were obtained from Japan SLC (Shizuoka, Japan) and kept in a temperature-controlled room (25˚C) with a 12:12 h light/dark cycle (lights were turned on at 8:00 a.m.) in the laboratory animal center of Osaka City University. Extensive care was taken during the experimental procedures to avoid causing any distress to the animals. After preliminary feeding for 8 days, the mice were randomly grouped so that the average weight of each group was equal, divided into the following six groups (6 mice/group), and fed for 52 days: ① control diet group (control), ② high fat diet group (HF), ③ high fat + 0.06% CPP (Light) diet group (HF + 0.06L), ④ high fat + 0.3% CPP (Light) diet group (HF + 0.3L), ⑤ high fat + 0.06% CPP (Extra Dark) diet group (HF + 0.06E), and ⑥ high fat + 0.3% CPP (Extra Dark) diet group (HF + 0.3E). Diet composition was prepared according to the AIN-93M (Table 1).

**Table 1. Diet composition.**

|  | Control | HF | HF + 0.06L | HF + 0.3L | HF + 0.06E | HF + 0.3E |
|---|---|---|---|---|---|---|
| Casein | 140.000 | 140.000 | 140.000 | 140.000 | 140.000 | 140.000 |
| Lard | 0 | 310.000 | 310.000 | 310.000 | 310.000 | 310.000 |
| L-Cystine | 1.800 | 1.800 | 1.800 | 1.800 | 1.800 | 1.800 |
| Cornstarch | 465.692 | 233.019 | 232.419 | 230.019 | 232.419 | 230.019 |
| α-cornstarch | 155.000 | 77.673 | 77.673 | 77.673 | 77.673 | 77.673 |
| Sucrose | 100.000 | 100.000 | 100.000 | 100.000 | 100.000 | 100.000 |
| Soybean Oil | 40.000 | 40.000 | 40.000 | 40.000 | 40.000 | 40.000 |
| Cellulose powder | 50.000 | 50.000 | 50.000 | 50.000 | 50.000 | 50.000 |
| AIN-93M mineral [a] | 35.000 | 35.000 | 35.000 | 35.000 | 35.000 | 35.000 |
| AIN-93 vitamin [b] | 10.000 | 10.000 | 10.000 | 10.000 | 10.000 | 10.000 |
| Choline Hydrogen Tartrate | 2.500 | 2.500 | 2.500 | 2.500 | 2.500 | 2.500 |
| tert-Butylhydroquinone | 0.008 | 0.008 | 0.008 | 0.008 | 0.008 | 0.008 |
| Carob extract (Light) | 0 | 0 | 0.600 | 3.000 | 0 | 0 |
| Carob extract (Extra Dark) | 0 | 0 | 0 | 0 | 0.600 | 3.000 |
| Total (g) | 1000 | 1000 | 1000 | 1000 | 1000 | 1000 |

[a] Composition in g/kg diet: Calcium carbonate, anhydrous, 357.00; potassium phosphate, monobasic, 250.00; sodium chloride, 74.00; potassium sulfate, 46.60; potassium citrate, 28.00; magnesium oxide, 24.00; ferric citrate, 6.06; zinc carbonate, 1.65; manganese carbonate, 0.63; cupric carbonate, 0.324; potassium iodate, 0.01; sodium selenate, anhydrous, 0.01025; ammonium paramolybdate, 4 hydrate, 0.00795; sodium meta-silicate, 9 hydrate, 1.45; chromium potassium sulfate, 12 hydrate, 0.275; lithium chloride, 0.0174; boric acid, 0.0815; sodium fluoride, 0.0635; nickel carbonate, 4 hydrate, 0.0306; ammonium vanadate, 0.0066; powdered sucrose, 209.7832.

[b] Composition in g/kg diet: nicotinic acid, 3.000; calcium pantothenate, 1.600; pyridoxine-HCl, 0.700; thiamin-HCl, 0.600; riboflavin, 0.600; folic acid, 0.200; d-biotin, 0.200; vitamin $B_{12}$ (cyanocobalamin) (0.l % in mannitol), 2.500; vitamin E (all-*rac*-α-tocopheryl acetate) (500 IU/g), 15.00; vitamin A (all-*trans*-retinyl palmitate) (500,000 IU/g), 0.800; vitamin $D_3$ (cholecalciferol) (400,000 IU/g), 0.250; vitamin K (phylloquinone), 0.075; Powdered sucrose, 974.655.

L(-)-Cystine, sucrose, choline hydrogen tartrate, and tert-butylhydroquinone were purchased from FUJIFILM Wako Pure Chemical Corporation, Japan. Casein, corn starch, αcorn starch, soybean oil, cellulose powder, AIN-93M mineral mix, and AIN-93 vitamin mix were purchased from Oriental Yeast Co., Ltd., Japan. Lard was added to the high fat diet. The estimated ratio of fat calories in the diet was 9.8% for the control diet and 59.4% for the high fat diet. Tap water was used as the drinking water. The mice were housed with three mice in one polycarbonate cage bearing stainless steel wire covers (CLEA Japan, Inc., Tokyo, Japan) with clean paper (Japan SLC, Shizuoka, Japan) and a polycarbonate igloo (Animec Ltd., Tokyo, Japan). The animals were handled by the same researcher. Food and water were freely available. Body weight and food intake were measured once every 2–3 days. Food consumption was measured for each cage, so no statistical analysis was performed. On the 52nd day of feeding, after a 5 h fast (free access to water only), the abdomen was opened under isoflurane inhalation anesthesia, and blood was collected from the abdominal vena cava. Thereafter, the liver, epididymal fat, retroabdominal fat, kidney, and spleen were removed. These organs were washed with physiological saline and weighed. The collected blood was centrifuged (3,500 rpm, 4˚C, 10 min) using a TOMY MX-160 centrifuge (TOMY SEIKO, Co., LTD., Japan) to obtain the serum. The obtained serum was stored at -80˚C until measurement.

## Blood biochemical analysis

Blood glucose was measured immediately after blood collection using the Glucose Pilot Blood Glucose Monitoring System. Serum triglyceride (TG) values were measured using a lab assay™ triglyceride (FUJIFILM Wako Pure Chemical Corporation) with the GPO/DAOS method. The serum total cholesterol level was measured using lab assay™ cholesterol (FUJIFILM Wako Pure Chemical Corporation) by the cholesterol oxidase/DAOS method.

## Histopathological analysis of liver and retroabdominal adipose tissue

The liver and retroabdominal adipose tissue were fixed with 10% neutral buffered formalin solution, and then paraffin specimens were prepared for hematoxylin and eosin (H&E) staining. Histopathological specimens were photographed using an inverted research microscope IX70 (Olympus Corporation, Japan) and the AdvanView imaging software. Adipocyte size was quantified using ImageJ software. The pathologist was blinded to the groups of mice.

## Cell culture

3T3-L1 preadipocytes (JCRB9014) were purchased from the Japanese Cancer Research Resources Bank and maintained in Dulbecco's modified Eagle's medium (DMEM) with 10% fetal bovine serum (FBS). At 2 days after reaching confluence (day 0), adipocyte differentiation was induced with 0.25 μM dexamethasone, 0.5 mM 3-isobutyl-1-methylxanthine, and 0.2 μM insulin (DMI) in DMEM supplemented with 10% FBS for 2 days. Then, the cells were treated with 0.2 μM insulin in DMEM supplemented with 10% FBS for another 2 days and cultured for an additional 4 days in DMEM with 10% FBS. CPP (Light) and CPP (Extra Dark) were dissolved in dimethyl sulfoxide (DMSO). The final DMSO concentration in the medium was < 0.5%. In all the experiments, control cultures were made up of medium, DMSO, and cells only.

## Cell viability measurement (neutral red method)

After culture with test agents, neutral red solution (0.25 mg/ml) was added to the culture at a final concentration of 50 μg/ml neutral red [12]. After incubation at 37˚C for 2 h, the cells

were rinsed twice with a solution of 1% (v/v) formaldehyde, 1% calcium chloride, and 98% distilled water. Then, the cells were destained with a buffer solution containing 1% (v/v) acetic acid, 50% (v/v) ethanol, and 49% (v/v) distilled water and incubated for 30 min. Lysosomal uptake of neutral red was quantified spectrophotometrically at 540 nm, and viability was calculated as follows:

$$\text{Viability (\%)} = \text{A540} - \text{treated cells}/\text{A540 of appropriate control} \times 100 \text{ after correction for background absorbance.}$$

## Measurement of TG accumulation

We evaluated the degree of differentiation into mature adipocytes, and staining was performed using Oil Red O [13]. The medium of 3T3-L1 adipocytes was removed with an aspirator and washed with 2 ml of $Ca^{++}$ and $Mg^{++}$ free-phosphate buffer saline (PBS (-)). Then, after fixing with 60% ethanol, 1 ml of Oil Red O staining solution was added and left at room temperature for 2 h. After washing with 50% ethanol, it was washed twice with 1 ml of ultrapure water and extracted with 1 ml of 2-propanol. The absorbance of the extract was measured at a wavelength of 520 nm using a spectrophotometer (JASCO V-730 BIO Spectrophotometer).

## Analysis of glycerol-3-phosphate dehydrogenase (GPDH) activity

3T3-L1 adipocytes were washed twice with 1 ml of PBS (-), and the cells were collected in 350 µl of triethanolamine/EDTA buffer with a cell scraper. Thereafter, the cells were sonicated using a sonicator (BIO RUPTOR, COSMO BIO Co., LTD., Japan). After centrifugation (13,000 rpm, 5 min, 4°C), the supernatant was used for the enzyme assay. The activity of GPDH was calculated using the extinction coefficient of nicotinamide adenine dinucleotide (NADH) of 6.22 $mM^{-1}cm^{-1}$ and was calculated based on the change in NADH per 3 min [14]. Enzyme activity was expressed as a value relative to the control (100%).

## Real-time PCR

Total RNA from 3T3-L1 adipocytes was extracted using a High Pure RNA Isolation Kit (Roche, Germany). Total RNA quality and quantity were evaluated using a 2100 Bioanalyzer (Agilent Technology, USA). cDNA was synthesized using the PrimeScript RT Reagent Kit (TaKaRa Bio Inc., Japan). Real-time PCR was performed using TB Green Premix Ex Taq II (TaKaRa Bio Inc.) according to the manufacturer's recommendations in a StepOnePlus PCR System (Thermo Fisher Scientific, USA). The primer sequences are shown in Table 2. The mRNA expression levels were normalized to β-actin. The StepOne software v2.2.2 (Thermo Fisher Scientific) was used for delta-delta CT analysis.

## Western blotting

3T3-L1 adipocytes were washed twice with 1 ml of PBS (-), and the cells were collected in 150 µl of RIPA buffer with a cell scraper. Thereafter, the cells were sonicated using a sonicator

**Table 2. The sequence of the primers.**

|  | Sense | Antisense |
|---|---|---|
| C/EBPβ | 5'-TTCTGTCTGTACGATTGTCAGTGGA-3' | 5'-GGCATGACTGGGCAGGATTA-3' |
| PPARγ | 5'-GGAGCCTAAGTTTGAGTTTGCTGTG-3' | 5'- TGCAGCAGGTTGTCTTGGATG-3' |
| C/EBPα | 5'- TTGAAGCACAATCGATCCATCC-3' | 5'-GCACACTGCCATTGCACAAG-3' |
| β-actin | 5'- CATCCGTAAAGACCTCTATGCCAAC-3' | 5- ATGGAGCCACCGATCCACA-3' |

(BIO RUPTOR, COSMO BIO Co., LTD., Japan). After centrifugation (15,000 rpm, 10 min, 4˚C), sample buffer was added to the supernatant and then heated at 90˚C for 5 min. The samples were stored at -80˚C before use. The protein concentration was assayed using the Pierce[TM] BCA protein assay kit (Thermo Fisher Scientific, USA). Total protein (20 μg) was separated by 10% SDS-PAGE and transferred to polyvinylidene difluoride membranes (Merck Millipore, USA). Anti-C/EBPβ(H-7) (Santa Cruz Biotechnology), anti-C/EBPα(14AA) (Santa Cruz Biotechnology), or anti-PPAR(E-8) (Santa Cruz Biotechnology) primary antibodies were diluted 1:2000. Furthermore, GAPDH (D16H11)XP rabbit mAb (Cell Signaling Technology, USA) or anti-actin antibody immunoglobulins/biotinylated (Dako, Denmark) as the loading control was diluted to 1:5000. After blocking with 3% bovine serum albumin for 1 h at room temperature, the membrane was incubated with primary antibody overnight at 4˚C and then incubated with a secondary antibody. For the secondary antibody, polyclonal goat anti-mouse immunoglobulins/biotinylated (Dako, Denmark) or polyclonal goat anti-rabbit immunoglobulins/biotinylated (Dako, Denmark) was diluted 1:3000. The membrane was visualized using EZ West Lumi One (ATTO Corporation, Japan) in AE-9300 EZ-Capture MG (ATTO). The fluorescence intensity was analyzed using the CS Analyzer ver. 3.0 (ATTO).

### Statistical analysis

Data are presented as mean ± SEM. Statistical analysis was performed using Statcel3, a useful add-in form, in Excel statistical software (OMS publishing Inc., Japan). For multiple comparisons, Dunnett's method was used in *in vivo* experiments, and the Tukey-Kramer method was used in *in vitro* experiments. A significant difference test was performed at a risk rate of 5% or 1%.

## Results

### Effect of CPPs on body weight

To determine the effect of a diet containing CPPs in C57BL/6J mice fed a high fat diet, we measured the body weights of the mice. The weight increased significantly in the HF group compared with the control group. However, there was a tendency for body weight to decrease in the CPP intake group with the HF group. In particular, in the HF + 0.06 L group, there was a significant decrease compared with the HF group (Fig 1).

### Effect of CPPs on visceral fat weight of mice

To examine whether the body weight gain in CPP-treated groups resulted in decreased fat accumulation, the visceral fat weight was examined. Visceral fat weight was significantly increased by a high fat diet. However, ingesting CPPs tended to decrease epididymal fat weight, and retroabdominal fat weight significantly decreased (Fig 2).

### Effect of CPPs on lipid metabolism of mice

To examine the effect of CPPs on lipid metabolism, we measured the serum total cholesterol levels of the mice. The serum total cholesterol level, which was significantly increased by the high fat diet, tended to decrease by feeding the mice with CPP (Light) (Fig 3). We also performed H&E staining of the liver and observed that in the HF group, lipid droplets accumulated in hepatocytes, and mice exhibited fatty liver. In all CPP-fed groups, fatty liver was remarkably suppressed (Fig 4A). These results were consistent with the liver TG levels (Fig 4B). These results suggest that CPPs inhibit fat accumulation in the liver induced by a high fat diet.

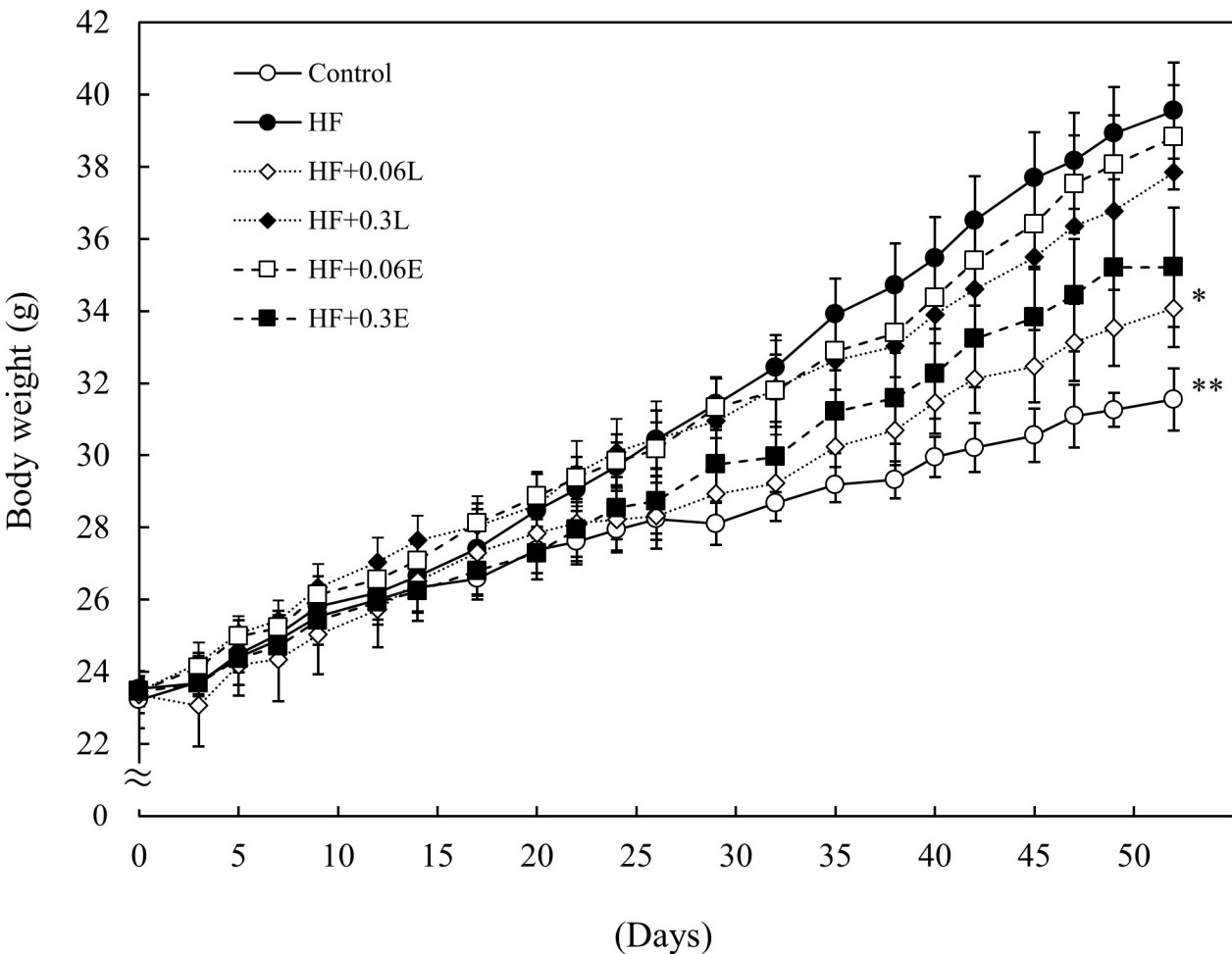

**Fig 1. Effect of CPP on body weight.** Control; Control food group, HF; High fat diet group, HF + 0.06L; High fat + 0.06% CPP (Light) diet group, HF + 0.3L; High fat + 0.3% CPP (Light) diet group, HF + 0.06E; High fat + 0.06% CPP (Extra Dark) diet group, HF + 0.3E; High fat + 0.3% CPP (Extra Dark) diet group. Values are means ± SEM of six mice. **; $p < 0.01$, *; $p < 0.05$ vs HF.

### Effect of CPPs on morphological changes in adipose tissue of mice

In the HF group, adipocyte hypertrophy was observed (Fig 5A). In addition, as shown by the arrow, many characteristic histological images associated with obesity were observed, in which macrophages surrounded the adipocytes that had died (crown-like structure, CLS). In the CPP-fed groups, macrophage infiltration was reduced. The intake of a high fat diet significantly increased the size of adipocytes. However, when CPPs were fed a high fat diet, adipocyte hypertrophy was significantly suppressed (Fig 5B).

### Effect of CPPs (Light and Extra Dark) treatment on the viability of 3T3-L1 preadipocytes

Cell viability of 3T3-L1 preadipocytes was examined with CPPs (Light and Extra Dark) by Neutral red assay. As shown in Fig 6A and 6B, neither type of CPP (Light or Extra Dark) had any effect on cell viability at concentrations of up to 100 μg/ml. These results showed that CPPs are not cytotoxic to 3T3-L1 preadipocytes.

(A)

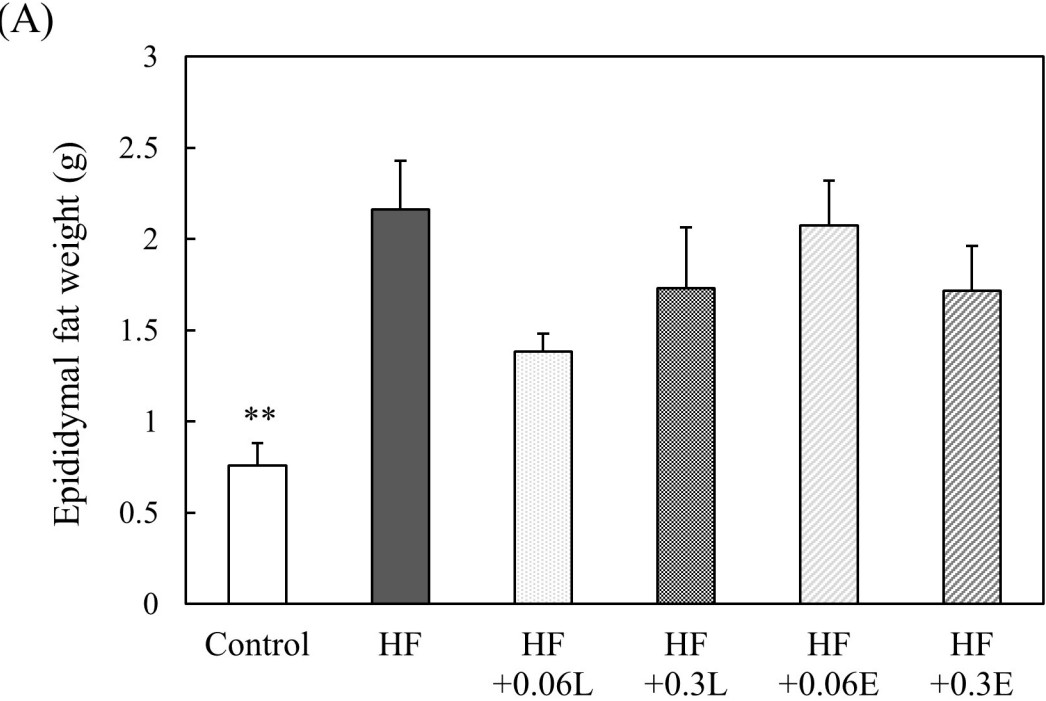

(B)

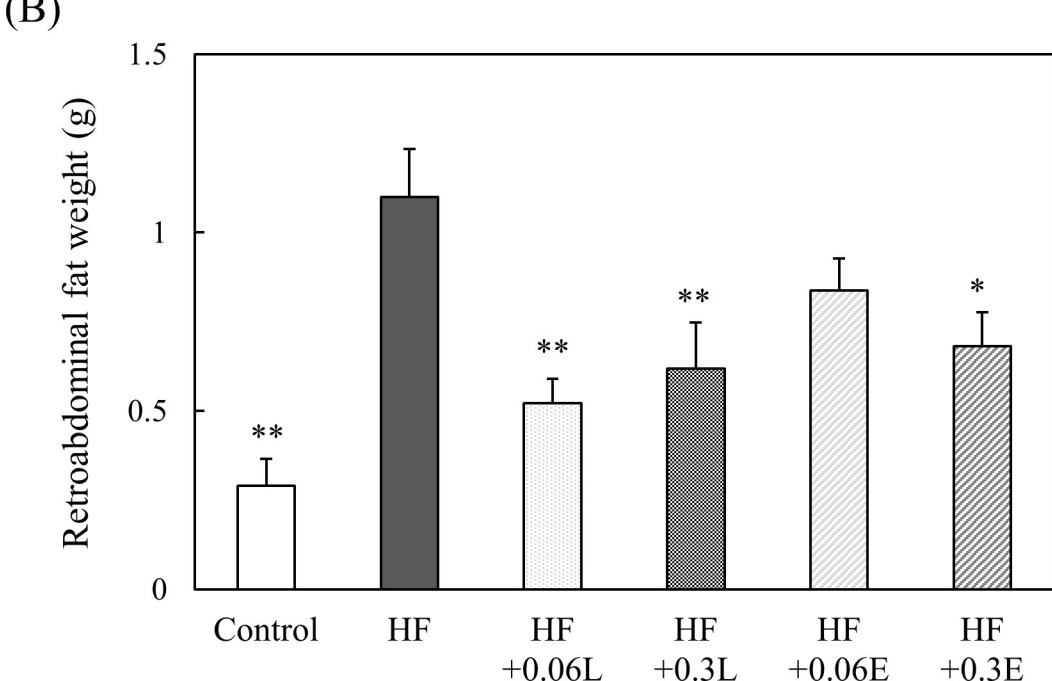

**Fig 2.** Effects of CPP on epididymal fat weight (A) and retroabdominal fat weight (B). Control; Control food group, HF; High fat diet group, HF + 0.06L; High fat + 0.06% CPP (Light) diet group, HF + 0.3L; High fat + 0.3% CPP (Light) diet group, HF + 0.06E; High fat + 0.06% CPP (Extra Dark) diet group, HF + 0.3E; High fat + 0.3% CPP (Extra Dark) diet group. Values are means ± SEM of six mice. **; $p < 0.01$, *; $p < 0.05$ vs HF.

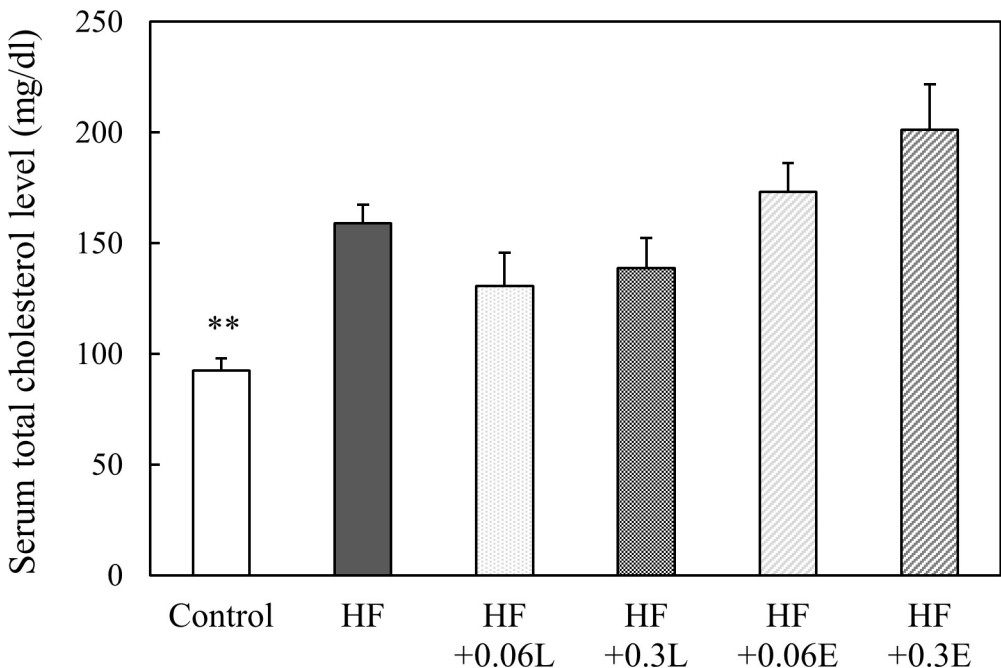

**Fig 3. Effect of CPP on the serum total cholesterol levels of mice.** The serum total cholesterol levels were measured by the cholesterol oxidase/DAOS method. Control; Control food group, HF; High fat diet group, HF + 0.06L; High fat + 0.06% CPP (Light) diet group, HF + 0.3L; High fat + 0.3% CPP (Light) diet group, HF + 0.06E; High fat + 0.06% CPP (Extra Dark) diet group, HF + 0.3E; High fat + 0.3% CPP (Extra Dark) diet group. Values are means ± SEM of six mice. **; $p < 0.01$ vs HF.

### Effect of CPP treatment on TG accumulation in 3T3-L1 preadipocytes

Intracellular TG was stained using the Oil Red O staining method. TG accumulation was significantly reduced in CPP (Light)-treated cells compared with CPP (Extra Dark)-treated cells. Furthermore, when CPP (Light) was added to 3T3-L1 preadipocytes at a concentration of 50 or 100 μg/ml, TG accumulation was significantly reduced, depending on the concentration of CPP (Light) (Fig 7A and 7B). Therefore, the following experiments were performed using CPP (Light).

### Effect of CPP (Light) on GPDH activity in 3T3-L1 preadipocytes

To ascertain the reduction of TG accumulation in 3T3-L1 preadipocytes treated with CPPs, we examined the effect of CPP (Light) on GPDH activity, a rate-limiting enzyme in TG synthesis. As shown in Fig 8, GPDH activity was significantly reduced, depending on the concentration of CPP (Light).

### Effect of the addition period of CPP (Light) on fat accumulation in 3T3-L1 preadipocytes

To clarify at which stage of differentiation that CPP (Light) is acting, an experiment was conducted by changing the sample addition period during the induction of differentiation (Fig 9A). When CPP (Light) was added on Day 0, intracellular TG accumulation decreased significantly. However, the addition of CPP (Light) after Day 2 had no effect on the intracellular TG level, showing that CPP (Light) acts at the early stages of adipocyte differentiation (Fig 9B).

(A)

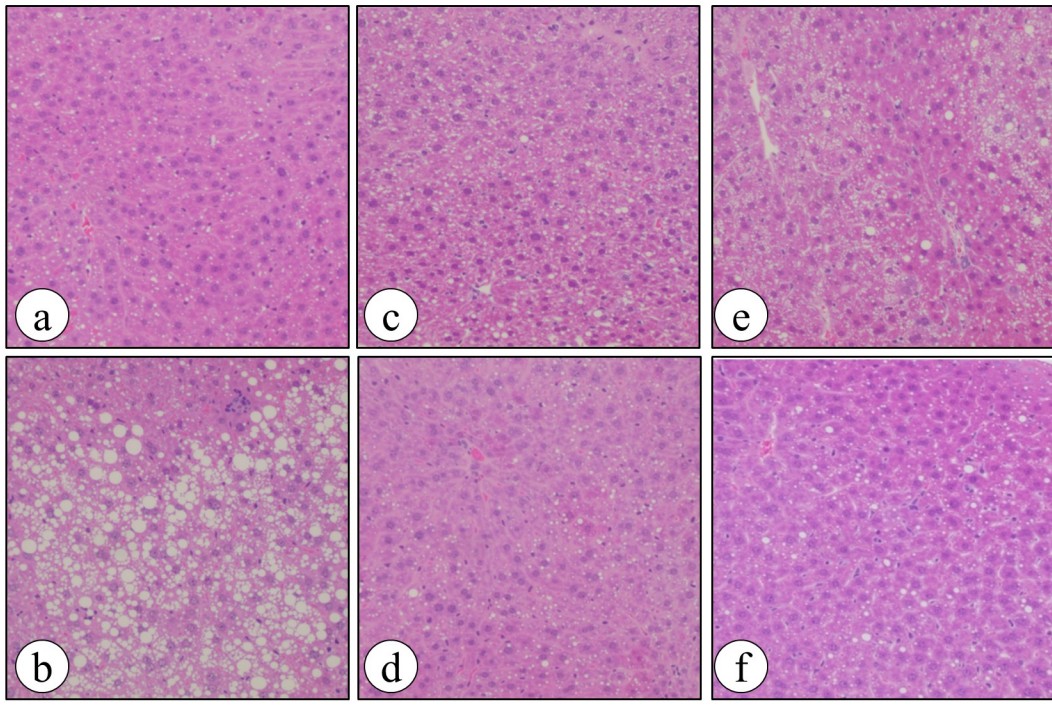

(B)

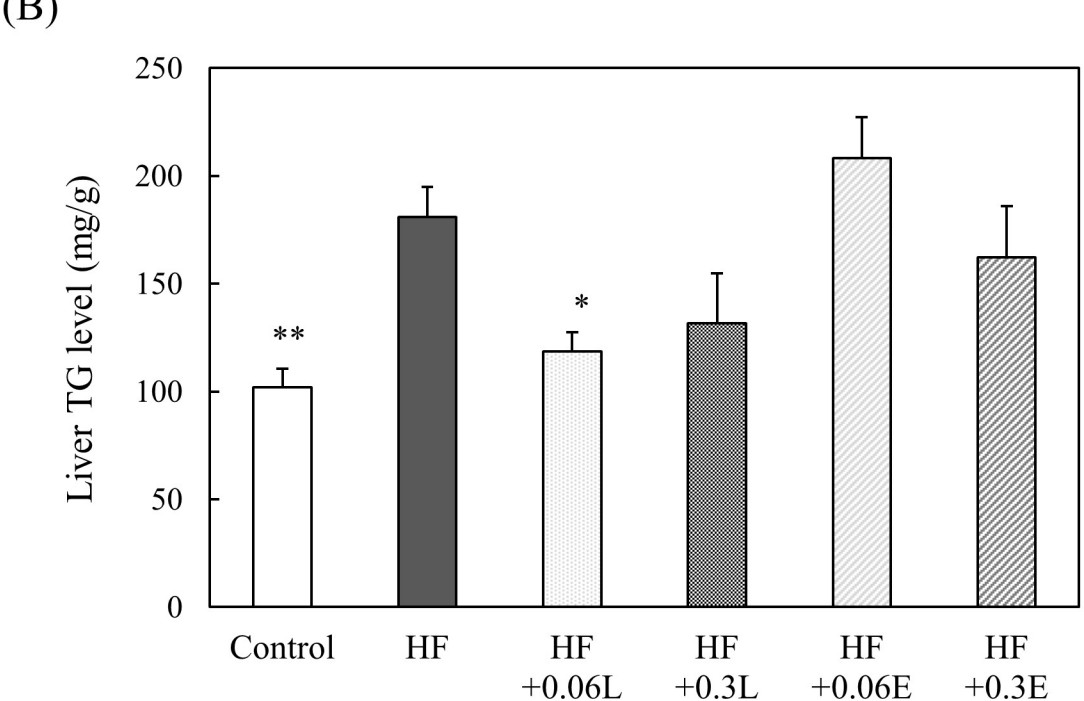

**Fig 4. Effects of CPP on morphological changes and TG levels in mouse livers.** (A) Liver specimens were stained with H&E staining. (a) Control food group, (b) High fat diet group, (c) High fat + 0.06% CPP (Light) diet group, (d) High fat + 0.3% CPP (Light) diet group, (e) High fat + 0.06% CPP (Extra Dark) diet group, (f) High fat + 0.3% CPP (Extra Dark) diet group. Original magnification: 20×. (B) Liver TG levels of mice were measured by the GPO/DAOS method. Control; Control food group, HF; High

fat diet group, HF + 0.06L; High fat + 0.06% CPP (Light) diet group, HF + 0.3L; High fat + 0.3% CPP (Light) diet group, HF + 0.06E; High fat + 0.06% CPP (Extra Dark) diet group, HF + 0.3E; High fat + 0.3% CPP (Extra Dark) diet group. Values are means ± SEM of six mice. **; $p < 0.01$, *; $p < 0.05$ vs HF.

### Effect of CPP (Light) treatment on mitotic clonal expansion (MCE) after inducing 3T3-L1 preadipocyte differentiation

Adipocytes stop growing as soon as they become confluent, but when stimulated to induce differentiation, the cell cycle progresses again and undergoes several MCEs to differentiate into adipocytes. Because CPP (Light) treatment acted in the early stage of differentiation, we thought that it might suppress MCE because of differentiation induction. Therefore, we examined the cell number by Trypan blue staining. Because of differentiation induction, MCE was significantly suppressed 48 h after the addition of carob extract (Fig 10).

### Effect of transcription factors involved in adipocyte differentiation in 3T3-L1 preadipocytes

Expression of the transcription factors involved in adipocyte differentiation was examined by real-time PCR for gene expression and western blotting for protein expression.

In adipocytes, the expression of C/EBPβ and C/EBPδ is induced in the early stage of differentiation, which is followed by the expression of PPARγ and C/EBPα, which are the master regulators of adipocyte differentiation.

The effect of CPP (Light) treatment on the expression of C/EBPβ was investigated. C/EBPβ is an important factor in MCE during early differentiation. At 48 h after the induction of differentiation, the gene expression levels of *C/EBPβ* did not change, even when CPP (Light) was added. However, the protein levels decreased significantly, depending on the concentration of CPP (Light) (Fig 11A and 11B). Therefore, it was revealed that the expression of C/EBPβ was regulated posttranscriptionally.

The effect of CPP treatment on the expression of PPARγ was investigated. CPP (Light) treatment significantly reduced *PPARγ* gene expression levels at both 24 and 48 h after the induction of differentiation (Fig 12A and 12B). By adding 100 μg/ml CPP (Light), PPARγ protein levels tended to decrease at 48 h and significantly decreased 72 h after the induction of differentiation (Fig 12C and 12D). Therefore, it was shown that CPP continuously reduced the *PPARγ* gene expression level starting at 24 h after the induction of differentiation and reduced the protein level after 72 h.

The effect of CPP (Light) treatment on the expression of C/EBPα was investigated. At 24 h after the induction of differentiation, the gene expression levels and protein levels of C/EBPα did not change, even when CPP (Light) was added (Fig 13A–13C). *C/EBPα* gene expression and protein levels at 48 h were significantly reduced by the addition of CPP (Light) (Fig 13B–13D). Therefore, the expression of C/EBPα was suppressed at 24–48 h after the induction of differentiation by adding CPPs.

## Discussion

The present study revealed that the carob extract significantly suppressed the increase in adipose tissue weight and adipocyte hypertrophy in the *in vivo* experimental system. Moreover, in the *in vitro* experimental system, CPP (Light) treatment acted at the early stages of differentiation of 3T3-L1 preadipocytes and suppressed cell proliferation because of differentiation induction. It suppressed the expression of transcription factors involved in adipocyte differentiation, reducing TG synthesis ability and TG accumulation. In addition, an interesting finding

(A)

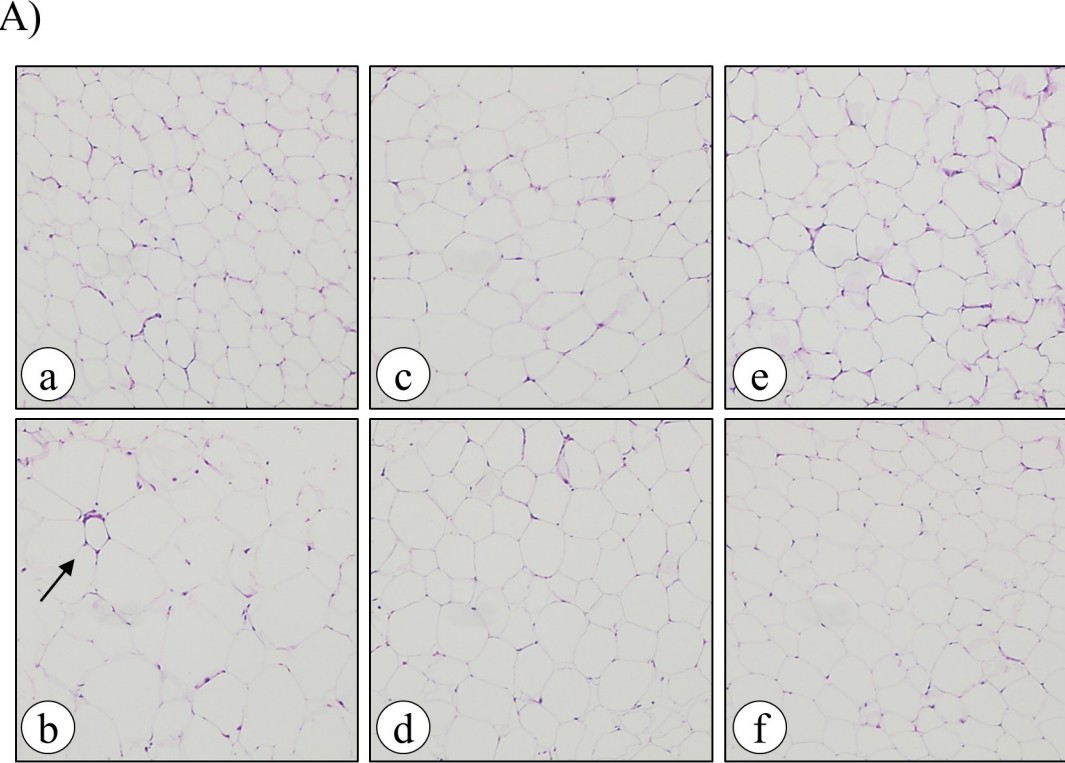

(B)

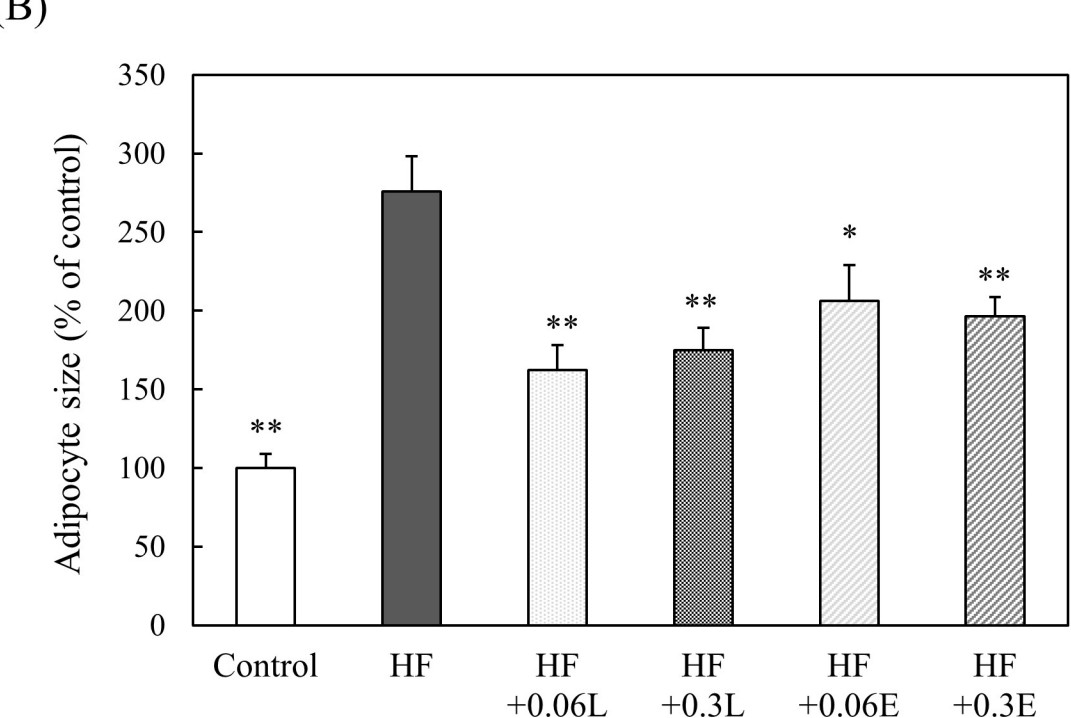

**Fig 5.** Effects of CPP on (A) morphological changes and (B) the size of adipocytes in adipose tissue of mice. (A) Retroabdominal adipose tissue specimens were stained with H&E. (a) Control food group, (b) High fat diet group, (c) High fat + 0.06% CPP (Light) diet group, (d) High fat + 0.3% CPP (Light) diet group, (e) High fat + 0.06% CPP (Extra Dark) diet group, (f) High fat

+ 0.3% CPP (Extra Dark) diet group. Original magnification: 10×. Arrow indicated a crown-like structure. (B) The size of adipocytes in retroabdominal adipose tissue specimens stained with H&E was quantified using ImageJ. Control; Control food group, HF; High fat diet group, HF + 0.06L; High fat + 0.06% CPP (Light) diet group, HF + 0.3L; High fat + 0.3% CPP (Light) diet group, HF + 0.06E; High fat + 0.06% CPP (Extra Dark) diet group, HF + 0.3E; High fat + 0.3% CPP (Extra Dark) diet group. Values are means ± SEM of six mice. **; p < 0.01, *; p < 0.05 vs HF.

is that CPPs treatment was found to regulate C/EBPβ expression at the early stage of differentiation at the posttranscriptional level.

In the *in vivo* experiment, we observed by histological analysis that the liver of the high fat diet group was a fatty liver. In the CPP-fed group, on the other hand, the occurrence of fatty liver was significantly suppressed (Fig 4A). Furthermore, the liver TG levels were also significantly increased by the high fat diet but decreased by feeding the mice CPPs (Fig 4B). These results suggest that CPPs inhibit fat accumulation in the liver induced by a high fat diet.

M1 macrophages, which increase with obesity, show a characteristic histology (crown-like structure, CLS) that surrounds adipocytes and secretes many inflammatory cytokines to promote inflammatory changes in adipose tissue [15]. In the present study, the histopathological analysis of retroperitoneal adipose tissue showed that fat cell hypertrophy and characteristic CLS were associated with obesity in the HF group (Fig 5A). However, when CPPs were fed a high fat diet, the hypertrophy of adipocytes was significantly suppressed (Fig 5B); the carob extract also reduced CLS. Therefore, CPPs were suggested to suppress fat cell hypertrophy and macrophage infiltration into fat cells induced by a high fat diet.

When CPP (Light) was added to 3T3-L1 preadipocytes at a concentration of 50 or 100 μg/ml, TG accumulation was significantly reduced, depending on the concentration of CPP

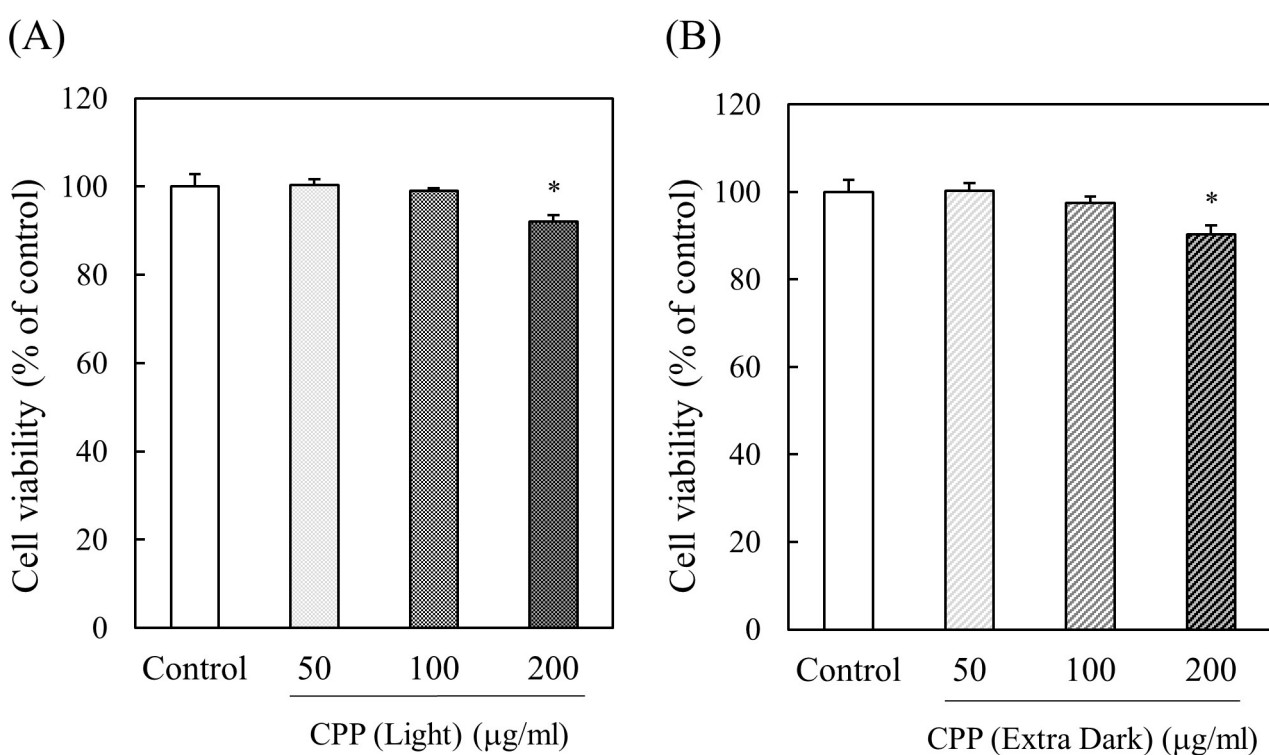

**Fig 6. Effect of CPPs (Light and Extra Dark) on cell viability of 3T3-Li preadipocytes.** Cells were cultured for 24 h with 50, 100, or 200 μg/ml CPPs (Light (A) and Extra Dark (B)). Cell viability was measured with Neutral red assay. Values are means ± SEM (n = 4). *; p < 0.05 vs Control.

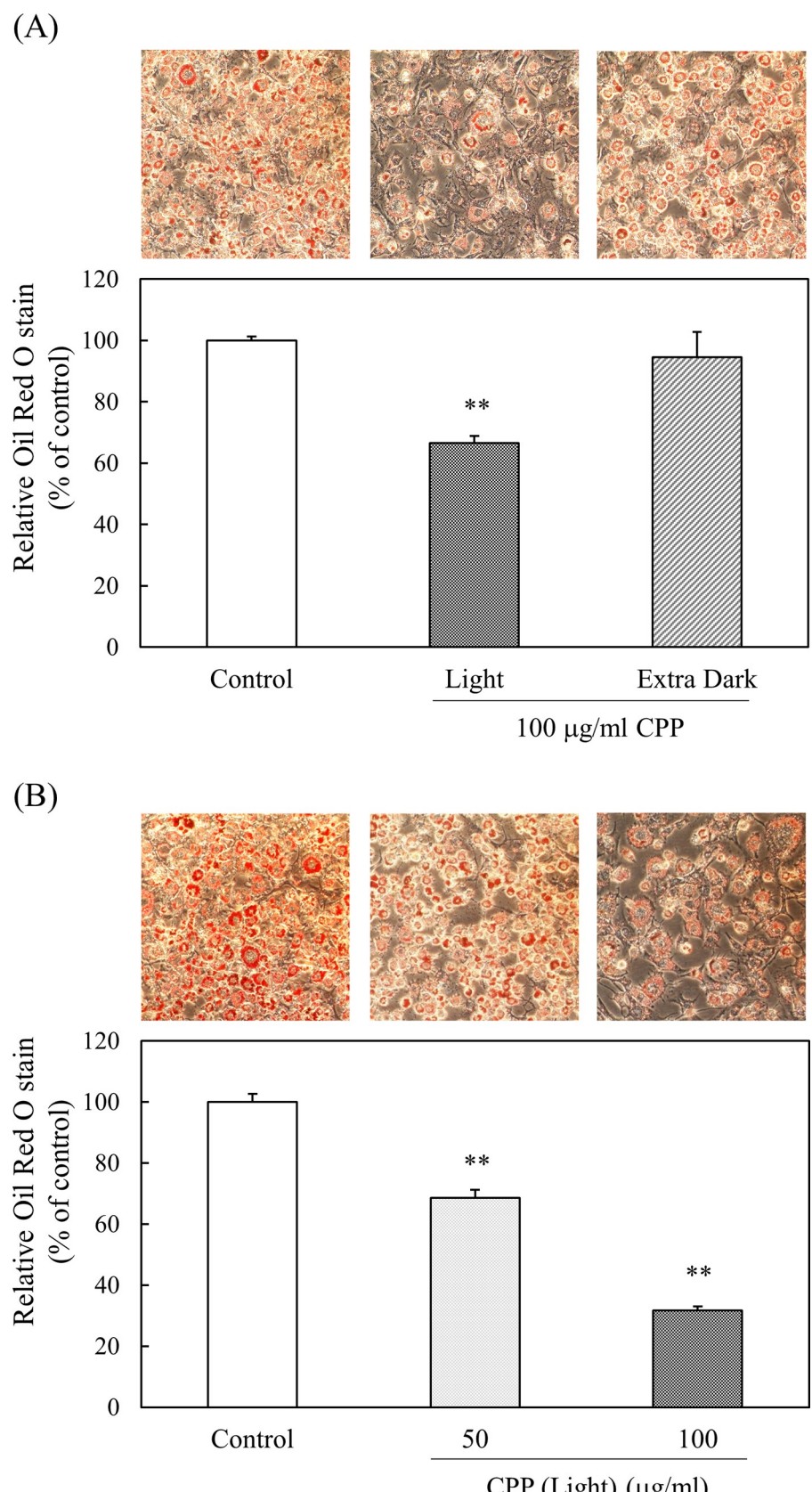

**Fig 7. Effect of CPP on TG accumulation of 3T3-L1 preadipocytes.** Intracellular TG was stained using the Oil Red O staining method. Oil Red O staining was performed 8 days after the initiation of differentiation. Representative photos of Oil Red O staining and quantitative analyses of 3T3-L1 preadipocytes treated with or without (A) 100 μg/ml CPP (Light or Extra Dark) or (B) 50 or 100 μg/ml CPP (Light). Oil Red O was extracted, and the absorbance of the extract was measured at a wavelength of 520 nm using a spectrophotometer. Values are means ± SEM (n = 4). **; p < 0.01 vs Control.

(Light) (Fig 7B). The addition of CPP (Light) to preadipocytes from the induction of differentiation reduced TG accumulation. These results suggest that CPPs have a "preventive" effect in obesity.

To clarify the stage at which differentiation CPP (Light) acts, we examined the sample addition period during the induction of differentiation. Only in the group to which CPP (Light) was added from the start of differentiation induction was intracellular TG accumulation significantly reduced. Therefore, it was revealed that CPP (Light) treatment acts at the early stage of adipocyte differentiation.

Adipocytes stop proliferating as soon as they become confluent, but upon receiving a differentiation-inducing stimulus, progress through the cell cycle starts again and undergoes several MCEs to differentiate into adipocytes [16]. Because CPP (Light) treatment acts at the early stage of differentiation, it might suppress MCE by the induction of differentiation; here, the cell number was examined by Trypan blue staining. MCE induced by differentiation was significantly suppressed 48 h after the addition of carob extract (Fig 10).

The expression of the transcription factors involved in adipocyte differentiation was examined by real-time PCR for gene expression and western blotting for protein expression. In adipocytes, the expression of C/EBPβ and C/EBPδ is induced in the early stage of differentiation,

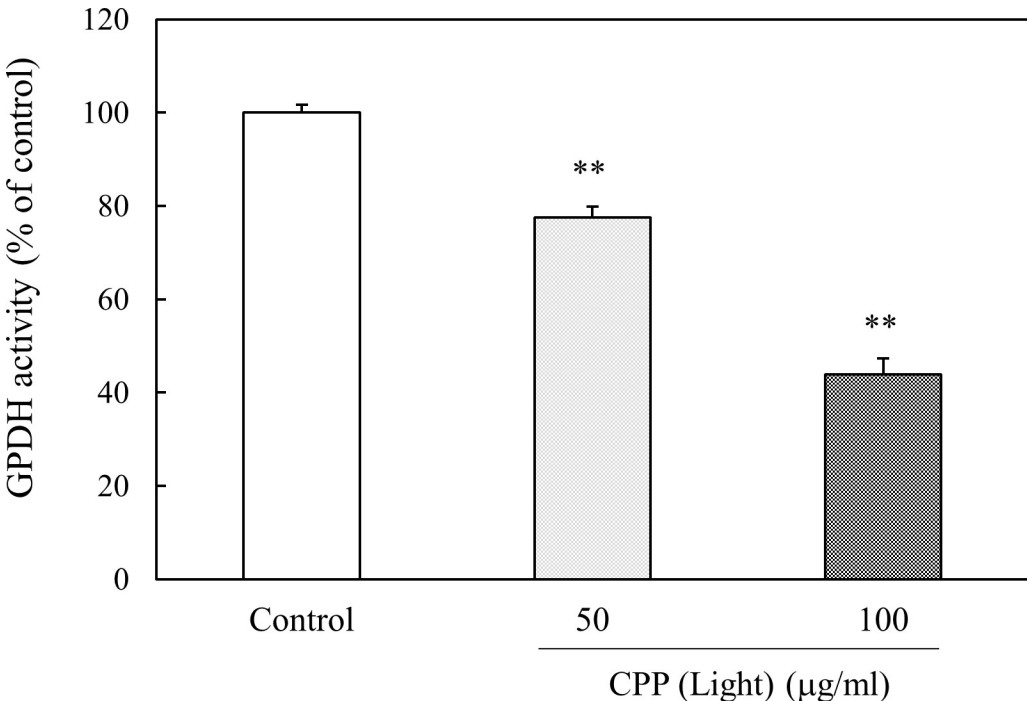

**Fig 8. Effect of CPP (Light) on GPDH activity in 3T3-L1 preadipocytes.** The activity of GPDH was calculated using the extinction coefficient of NADH, 6.22 mM⁻1 cm⁻¹, and was calculated based on the decrease in NADH every 3 min. Enzyme activity is expressed as a value relative to the control (100%). Values are means ± SEM (n = 4). **; p < 0.01 vs Control.

(A)

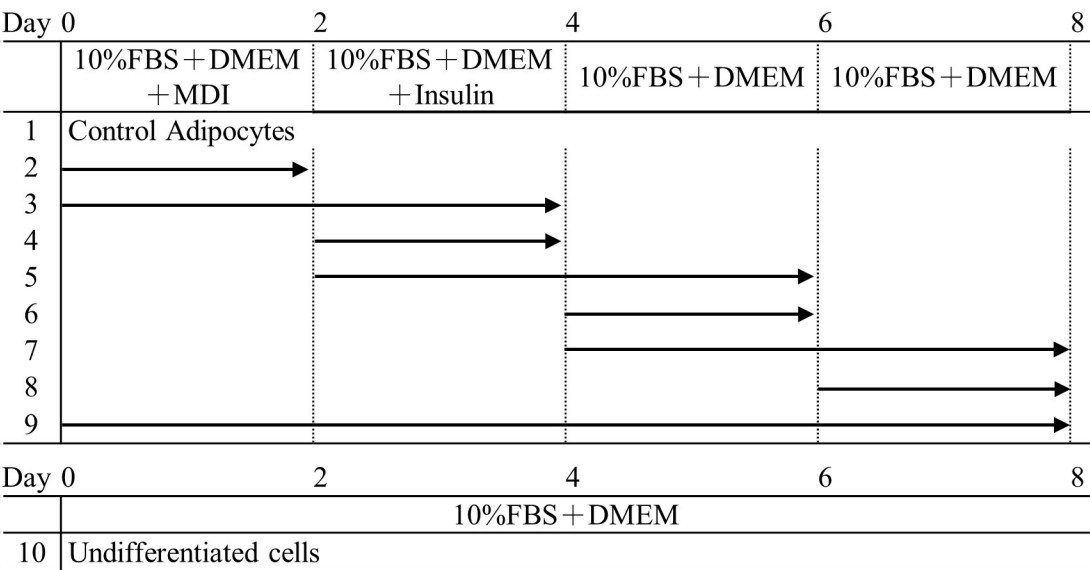

(B)

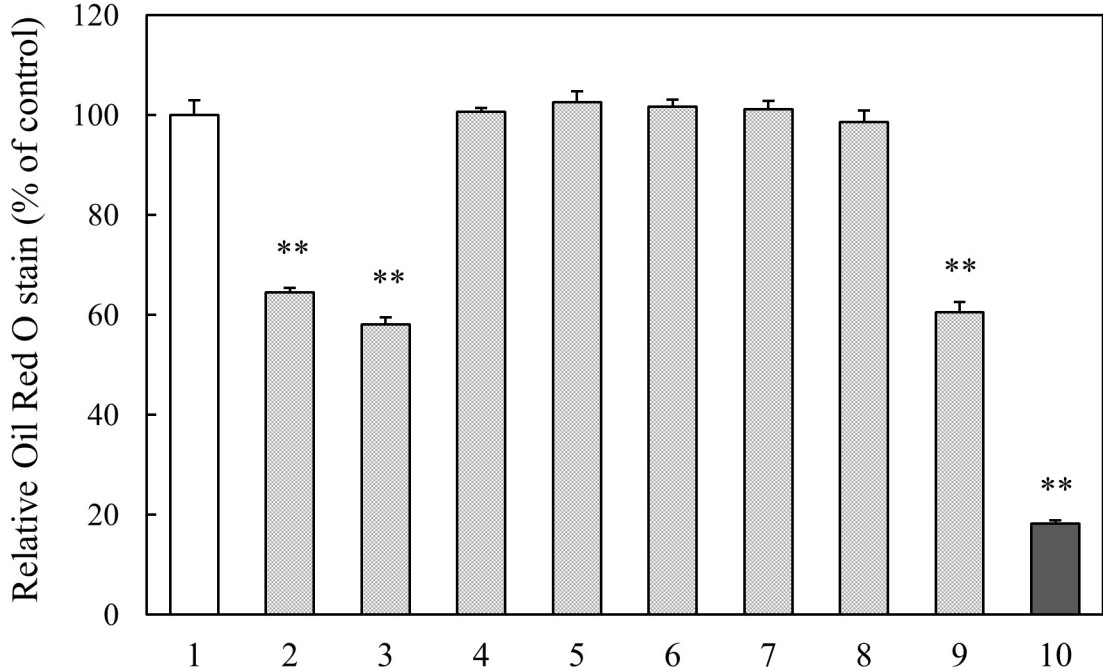

**Fig 9. Effect of the addition period of CPP (Light) on fat accumulation in 3T3-L1 preadipocytes.** (A) The schema shows the addition of CPP (Light) during the induction of differentiation. Oil Red O staining was performed 8 days after the initiation of differentiation. (B) Oil Red O was extracted, and the absorbance of the extract was measured at a wavelength of 520 nm using a spectrophotometer. Values are means ± SEM (n = 4). **; $p < 0.01$ vs Control.

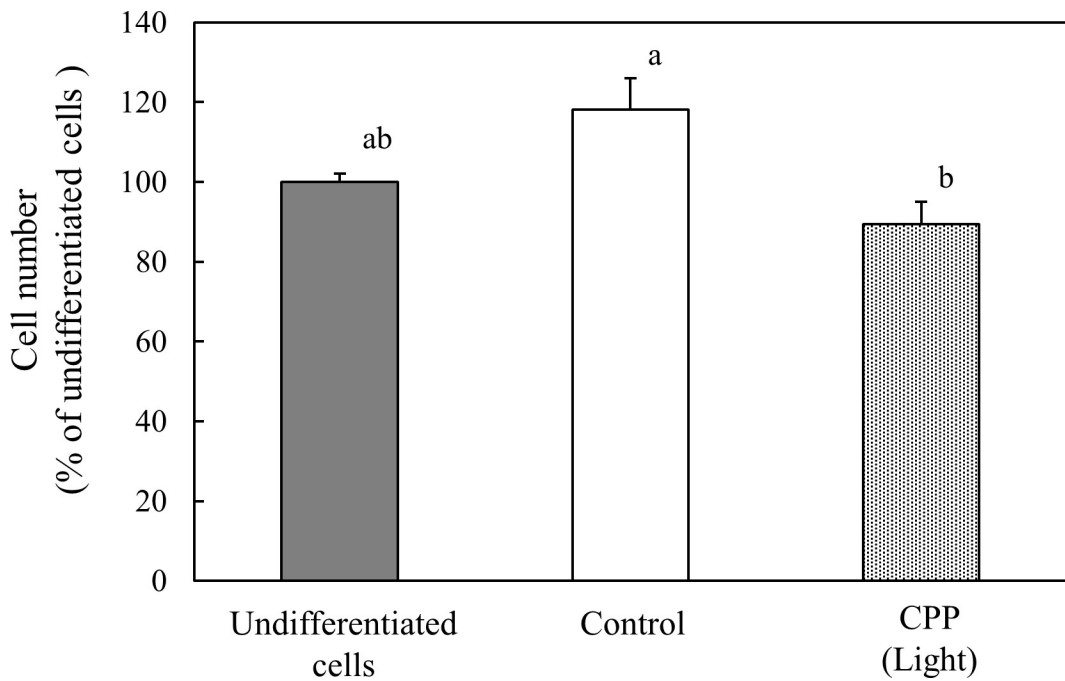

**Fig 10. Effect of CPP (Light) on mitotic clonal expansions after inducing 3T3-L1 preadipocyte differentiation.** The cell number was measured with Trypan blue staining at 48 h after inducing 3T3-L1 preadipocyte differentiation. Values are means ± SEM (n = 3). Values without a common letter are significantly different (p < 0.05).

and gene expression reaches a peak 1–2 days after the induction of differentiation [17]. These transcription factors induce the expression of PPARγ and C/EBPα, which are the master regulators of adipocyte differentiation [18, 19]. PPARγ and C/EBPα do not function independently but maintain their expression in cooperation with each other and induce the expression of adipocyte differentiation markers [20].

First, the effect of CPP treatment on C/EBPβ expression was examined. C/EBPβ is an important factor in MCE during early differentiation [21]. Indeed, 48 h after the induction of differentiation, the gene expression level of *C/EBPβ* did not change when CPP (Light) was added (Fig 11A). On the other hand, the protein levels decreased significantly, depending on the concentration of CPP (Light) (Fig 11B). Therefore, the expression of C/EBPβ appeared to be regulated post-transcriptionally.

Next, the effect of CPP treatment on the expression of PPARγ was examined. CPP (Light) treatment significantly reduced the level of *PPARγ* gene expression 24 h (Fig 12A) and 48 h (Fig 12B) after induction of differentiation. By adding CPP (Light) at 100 μg/ml, PPARγ protein levels tended to decrease 48 h after induction of differentiation (Fig 12C) and significantly decreased after 72 h (Fig 12D). Therefore, it was shown that CPPs continuously reduced the *PPARγ* gene expression levels from 24 h after the induction of differentiation and reduced the protein levels after 72 h. Furthermore, the effect of CPP (Light) treatment on the expression of C/EBPα was examined. At 24 h after the induction of differentiation, the gene expression levels of *C/EBPα* (Fig 13A) and protein levels (Fig 13C) were not changed by the addition of CPP (Light). At 48 h, the levels of *C/EBPα* gene expression (Fig 13B) and protein levels (Fig 13D) were significantly reduced by the addition of CPP (Light). Therefore, the expression of C/EBPα was suppressed 24 to 48 h after the induction of differentiation by adding CPPs.

The effect of CPP (Light) treatment on the expression of a transcription factor (C/EBPβ, PPARγ, and C/EBPα) involved in adipocyte differentiation is shown in Scheme 1.

(A)

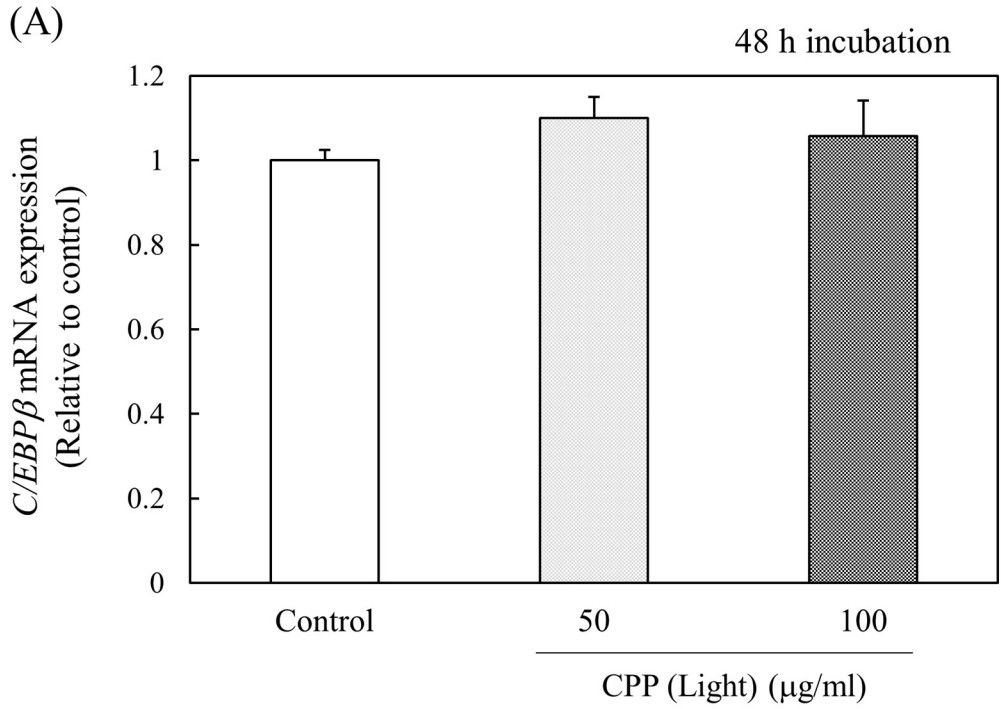

(B)

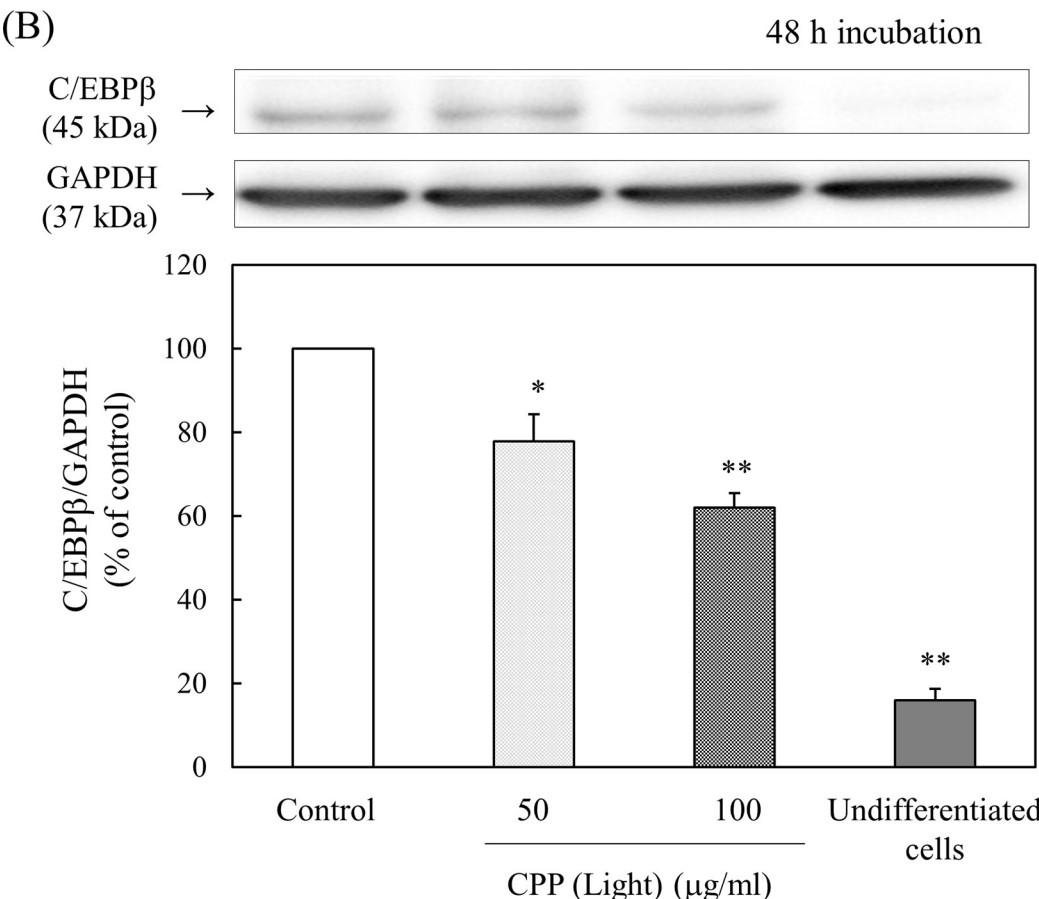

**Fig 11. Effects of CPP (Light) on (A) the gene expression and (B) protein levels of C/EBPβ.** (A) mRNA expression of *C/EBPβ* was analyzed with real-time PCR 48 h after induction of 3T3-L1 preadipocyte differentiation. Values are means ± SEM (n = 4). (B) C/EBPβ; protein levels were measured with western blotting 48 h after the induction of 3T3-L1 preadipocyte differentiation. Values are means ± SEM (n = 3). **; p < 0.01, *; p < 0.05 vs Control.

C/EBPβ was posttranscriptionally regulated by CPP (Light) treatment. It has been reported that the stability of C/EBPβ protein is increased by oxidative modification (S-glutathionization). In 3T3-L1 cells, glutaredoxin, an enzyme that catalyzes de-S-glutathionase, was knocked out, S-glutathionisation increased, fat synthesis was promoted, and PPARγ, C/EBPα, and C/EBPβ expression increased. It has been suggested that C/EBPβ expression is regulated at the protein level [22]. CPPs have been reported to have antioxidant activity, indicating that in the current study, CPP (Light) treatment inhibited the oxidative modification of C/EBPβ protein, resulting in C/EBPβ not being able to acquire stability and degradation.

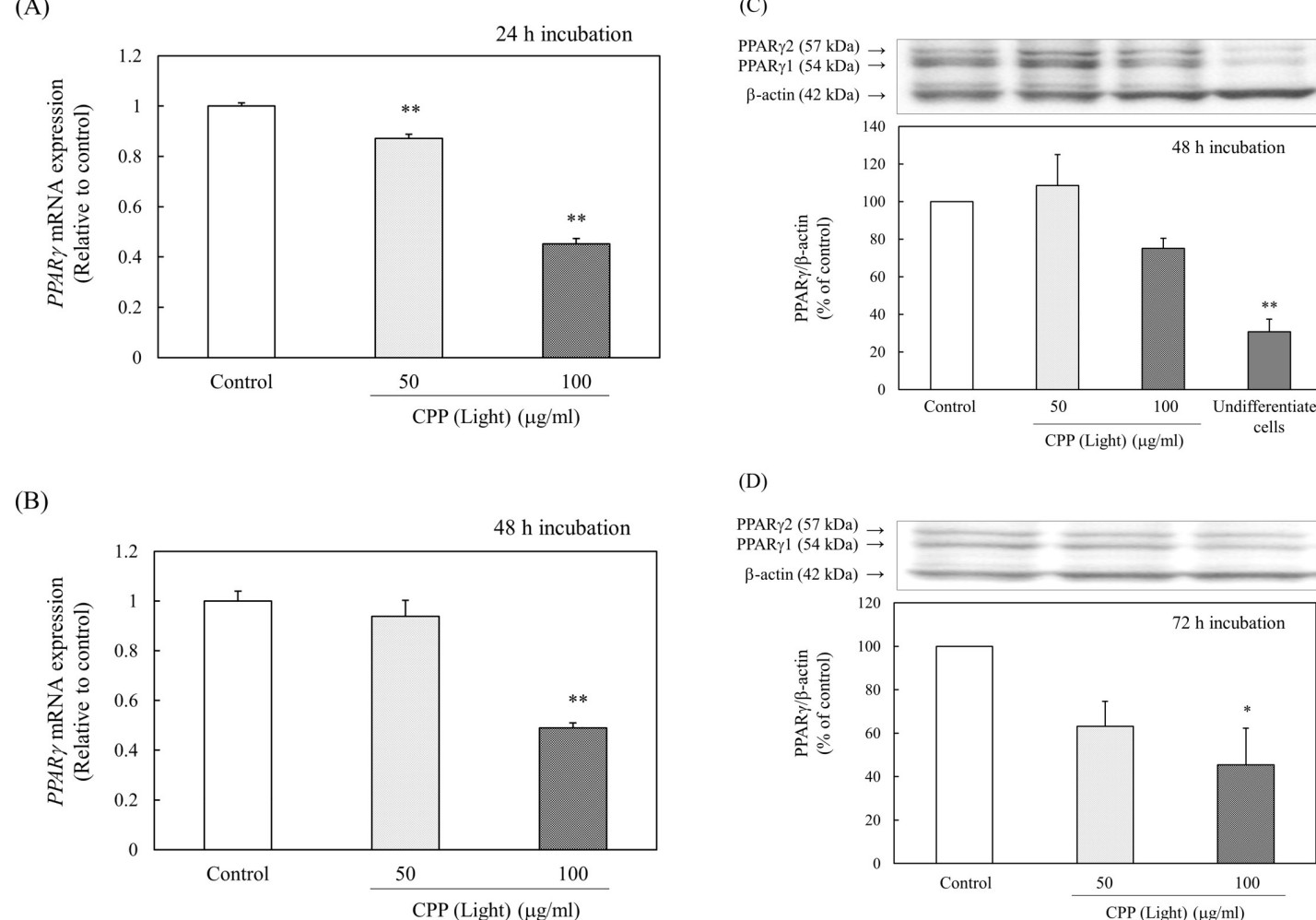

**Fig 12.** Effects of CPP (Light) on (A, B) the gene expression and (C, D) protein levels of PPARγ. (A, B) mRNA expression of *PPARγ* was analyzed with real-time PCR 24 h (A) or 48 h (B) after induction of 3T3-L1 preadipocyte differentiation. Values are means ± SEM (n = 4). **; p < 0.01 vs Control. (C, D) PPARγ protein levels were measured with western blotting 48 h (C) or 72 h (D) after induction of 3T3-L1 preadipocyte differentiation. Values are means ± SEM (n = 3). **; p < 0.01, *; p < 0.05 vs Control.

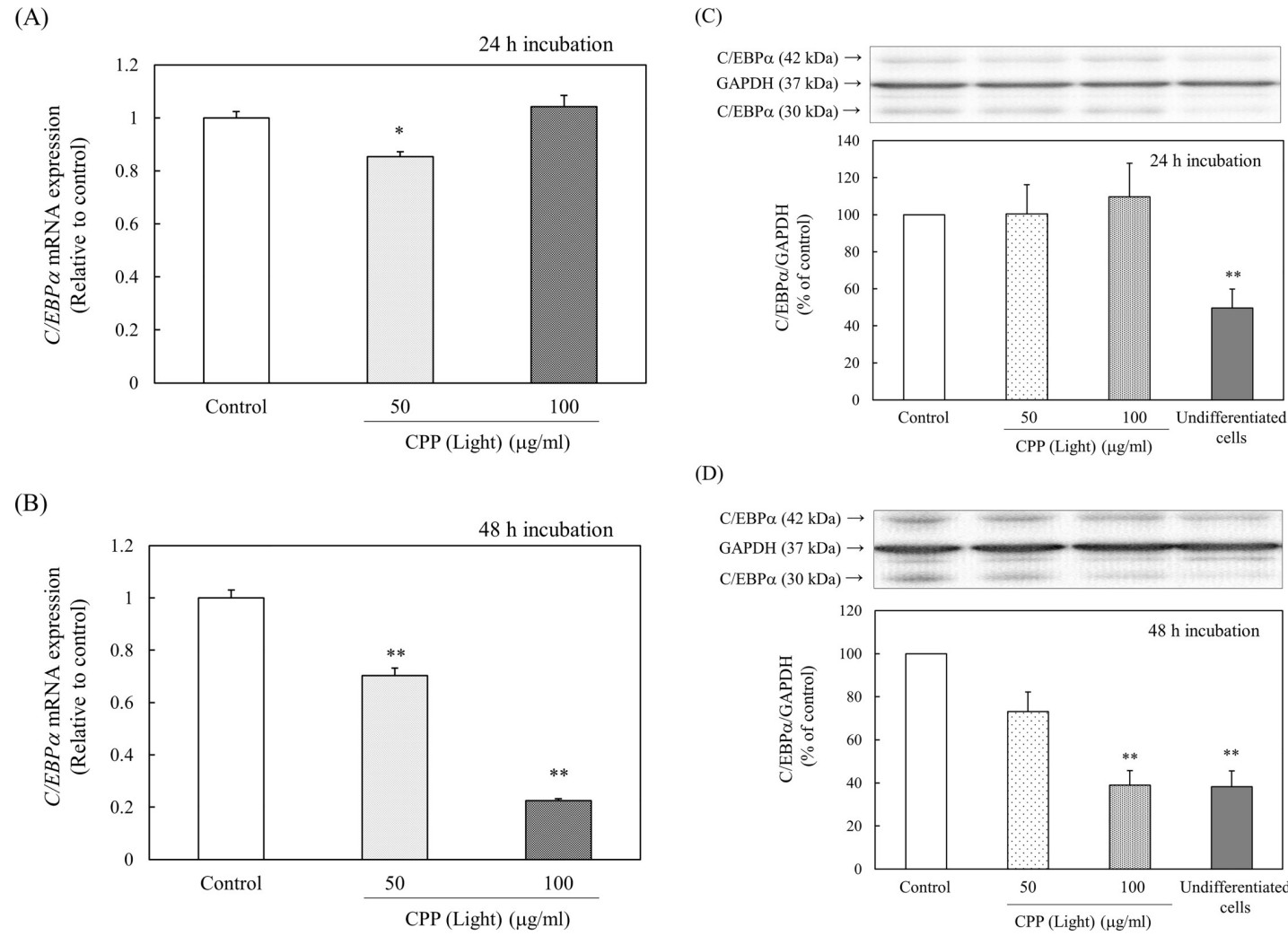

**Fig 13.** Effects of CPP (Light) on (A, B) the gene expression and (C, D) protein levels of C/EBPα. (A, B) mRNA expression of *C/EBPα* was analyzed with real-time PCR 24 h (A) or 48 h (B) after induction of 3T3-L1 preadipocyte differentiation. Values are means ± SEM (n = 4). **; $p < 0.01$, *; $p < 0.05$ vs Control. (C, D) C/EBPα protein levels were measured with western blotting 24 h (C) or 48 h (D) after induction of 3T3-L1 preadipocyte differentiation. Values are means ± SEM (n = 3). **; $p < 0.01$ vs Control.

Finally, we also wanted to elucidate the active substance of the carob extract, which has an anti-obesity effect. Carob pods are rich in polyphenols (tannins, flavonoids, and phenolic acids) [23]. Carobs have been reported to reduce hydrolyzable tannins and increase gallic acid through the process of roasting [24]. In preliminary experiments, we found in studies using 3T3-L1 preadipocytes that gallic acid has no anti-obesity effect (unpublished data). Therefore, it is suggested that CPP (Extra Dark) carob pods roasted for a long time had a lower anti-obesity effect than CPP (Light) due to a decrease in unknown components having an anti-obesity effect and an increase in gallic acid, which had no anti-obesity effect. In addition, catechins form tannins through oxidation and polymerization. This suggests that CPP (Extra Dark) was not as effective as CPP (Light) because the polymerization of polyphenols progressed during roasting. It is thought that the combination of components, such as the proanthocyanidin-based polymerized catechins contained in CPPs, exerts an anti-obesity effect, but isolating the

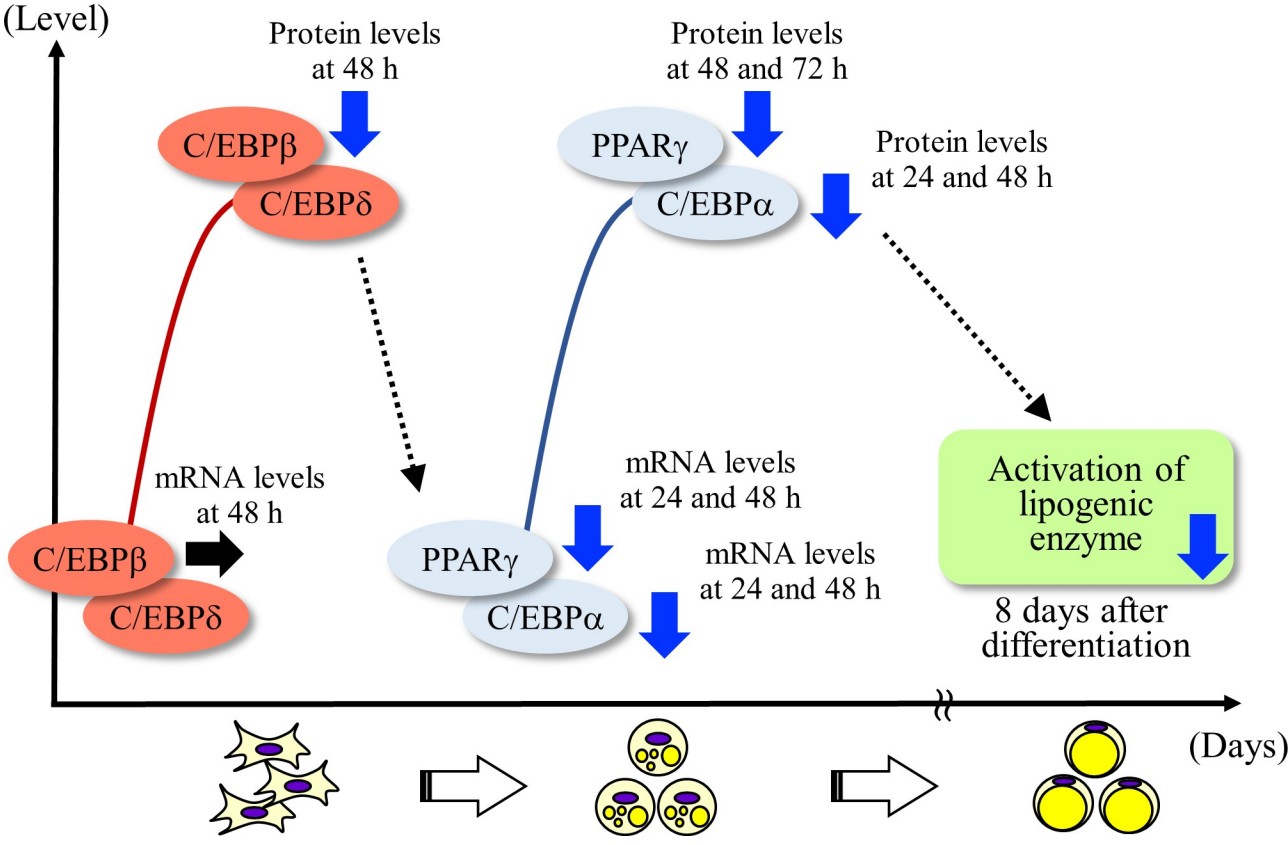

**Scheme 1. Effect of CPP on the expression of transcription factors in the differentiation of 3T3-L1 preadipocytes.** →; no effect, ↓; down regulation.

active substance for identification is currently difficult. It is necessary to further study the isolation and purification of polymerized polyphenols.

In conclusion, these results demonstrated that CPPs suppressed the differentiation of adipocytes through the posttranscriptional regulation of C/EBPβ and may be an effective anti-obesity compound.

## Supporting information

**S1 Raw images. Original images of Figs 11B, 12C, 12D, 13C and 13D.**
(PDF)

## Author Contributions

**Investigation:** Kasumi Fujita, Toshio Norikura, Shigenori Kumazawa, Sari Honda, Takumi Sonoda, Akiko Kojima-Yuasa.

**Methodology:** Toshio Norikura, Isao Matsui-Yuasa.

**Supervision:** Akiko Kojima-Yuasa.

**Writing – original draft:** Kasumi Fujita, Isao Matsui-Yuasa, Akiko Kojima-Yuasa.

**Writing – review & editing:** Isao Matsui-Yuasa, Akiko Kojima-Yuasa.

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
