## [Decision Letter · Decision Letter 0]

25 Jan 2021

PONE-D-20-38421

Carob pod polyphenols suppress the differentiation of adipocytes through posttranscriptional regulation of C/EBPß

PLOS ONE

Dear Dr. Kojima-Yuasa,

Thank you for submitting your manuscript to PLOS ONE. After careful consideration, we feel that it has merit but does not fully meet PLOS ONE’s publication criteria as it currently stands. Therefore, we invite you to submit a revised version of the manuscript that addresses the points raised during the review process.

We look forward to receiving your revised manuscript.

Kind regards,

Juan J Loor

Academic Editor

PLOS ONE

Journal Requirements:

We note that one or more of the authors are employed by a commercial company: TAISHO TECHNOS, Co., Ltd.

2.1. Please provide an amended Funding Statement declaring this commercial affiliation, as well as a statement regarding the Role of Funders in your study. If the funding organization did not play a role in the study design, data collection and analysis, decision to publish, or preparation of the manuscript and only provided financial support in the form of authors' salaries and/or research materials, please review your statements relating to the author contributions, and ensure you have specifically and accurately indicated the role(s) that these authors had in your study. You can update author roles in the Author Contributions section of the online submission form.

2.2. Please also provide an updated Competing Interests Statement declaring this commercial affiliation along with any other relevant declarations relating to employment, consultancy, patents, products in development, or marketed products, etc.  

5. Please include a caption for figures 13A-D.

6. Please ensure that you refer to Figures 13A-D in your text as, if accepted, production will need this reference to link the reader to the figure.

Reviewers' comments:

Reviewer's Responses to Questions

**Comments to the Author**

1. Is the manuscript technically sound, and do the data support the conclusions?

Reviewer #1: Yes

2. Has the statistical analysis been performed appropriately and rigorously? 

Reviewer #1: Yes

3. Have the authors made all data underlying the findings in their manuscript fully available?

Reviewer #1: Yes

4. Is the manuscript presented in an intelligible fashion and written in standard English?

Reviewer #1: No

5. Review Comments to the Author

Reviewer #1: There is a major problem in lining up your figures/data with the descriptions in the text. For example on line 294, it states that Fig 7 displays the effect of CPP (Light) on GPDH activity. However, in the text figures these enzyme activity results are in Fig 8. The problem appears to be that the data for CPP treatment on TG accumulation was to be Fig 6, but in the MS Fig 6 is about adiponectin and the data are missing. From that point on all figures are off, ie Fig 6 is really Fig 7 etc. These presentation errors must be corrected. Somehow the adiponectin results seem to have been deleted from the MS, but the Figure #s were never adjusted.

Overall the presentation appears somewhat "unconcise", this reviewer did not like a short results description in the text and this was followed by the legend of the Fig for those results in the complete MS.

Fortunately the discussion does enlighten the readers and I believe the research work was done properly. But again the layout of the MS requires work .

6. PLOS authors have the option to publish the peer review history of their article (what does this mean?). If published, this will include your full peer review and any attached files.

Reviewer #1: No

---

## [Author Response · Author response to Decision Letter 0]

13 Feb 2021

RESPONSES TO COMMENTS BY THE ACADEMIC EDITOR AND REVIEWER #1 

(PONE-D-20-38421)

We would like to thank your valuable comments for the helpful constructive comments and suggestions on our manuscript entitled “Carob pod polyphenols suppress the differentiation of adipocytes through posttranscriptional regulation of C/EBP�”. 

Below are our responses to the concerns raised by the Academic editor and Reviewer #1. There reviews were thorough and we are grateful for their contributions and have addressed each one of these. 

We sincerely believe our manuscript is now much improved. 

Responses to “Journal Requirements”:

 → We checked our manuscript according to PLOS ONE’s style.

2.1. Please provide an amended Funding Statement declaring this commercial affiliation, as well as a statement regarding the Role of Funders in your study. If the funding organization did not play a role in the study design, data collection and analysis, decision to publish, or preparation of the manuscript and only provided financial support in the form of authors' salaries and/or research materials, please review your statements relating to the author contributions, and ensure you have specifically and accurately indicated the role(s) that these authors had in your study. You can update author roles in the Author Contributions section of the online submission form.

Please also include the following statement within your amended Funding Statement. “The funder provided support in the form of salaries for authors [insert relevant initials], but did not have any additional role in the study design, data collection and analysis, decision to publish, or preparation of the manuscript. The specific roles of these authors are articulated in the ‘author contributions’ section.” If your commercial affiliation did play a role in your study, please state and explain this role within your updated Funding Statement.

 → Thank you very much for your kindly comments and valuable suggestions. We added the Funding and Competing interest as following in red characters (Page 28, Line 517 to 528).

Funding

 This work was supported by JSPS KAKENHI (Grant Number JP15K00832). KAKENHI aims to significantly develop all "academic research" from basic to applied (research based on the free thinking of researchers). TAISHO TECHNOS, Co., Ltd. provided the Carob pod polyphenols, a part of the grant, and support in the form of salary for T.S., but did not have any additional role in the study design data collection and analysis decision to publish, or preparation of the manuscript. The specific role of T.S. is articulated in the “Author Contributions” section.

Competing interests

 A part of research grant and Carob pod polyphenols were provided by the TAISHO TECHNOS, Co., Ltd. We declare that these relationships did not affect the results and conclusions of this manuscript. This does not alter on adherence to PLOS ONE politics on sharing data and materials.

2.2. Please also provide an updated Competing Interests Statement declaring this commercial affiliation along with any other relevant declarations relating to employment, consultancy, patents, products in development, or marketed products, etc.  Within your Competing Interests Statement, please confirm that this commercial affiliation does not alter your adherence to all PLOS ONE policies on sharing data and materials by including the following statement: "This does not alter our adherence to PLOS ONE policies on sharing data and materials.” 

If this adherence statement is not accurate and  there are restrictions on sharing of data and/or materials, please state these. Please note that we cannot proceed with consideration of your article until this information has been declared.

Please include both an updated Funding Statement and Competing Interests Statement in your cover letter. 

 → Thank you very much for your valuable suggestions. We added the Competing interest as following in red characters (Page 28, Line 525 to 528).

Competing interests

 A part of research grant and Carob pod polyphenols were provided by the TAISHO TECHNOS, Co., Ltd. We declare that these relationships did not affect the results and conclusions of this manuscript. This does not alter on adherence to PLOS ONE politics on sharing data and materials.

 And we included both an updated Funding Statement and Competing Interests Statement in our cover letter.

3. When you submit your revised manuscript, please ensure that your figures adhere fully to these guidelines and provide the original underlying images for all blot or gel data reported in your submission. 

In your cover letter, please note whether your blot/gel image data are in Supporting Information or posted at a public data repository.

 → Thank you very much for your valuable suggestions. We added the three supplemental files of the original underlying images for all blot. And we wrote about it in our cover letter.

 → Thank you very much for your kindly comments and valuable suggestions. We have included 3 phrases “data not shown” in our manuscript.

1) The data of organ weight are not a core part of our research, so we changed as following, 

 Effect of CPPs on organ weight of mice

 There was no significant difference in the liver, kidney, or spleen weights in each group (data not shown). Visceral fat weight was significantly increased by a high-fat diet. However, ingesting CPPs tended to decrease epididymal fat weight, and retroabdominal fat weight significantly decreased (Fig. 2).

　　　　　　　　　　　　　 ↓

 Effect of CPPs on visceral fat weight of mice

 To examine whether the body weight gain in CPP-treated groups resulted in decreased fat accumulation, the visceral fat weight was examined. Visceral fat weight was significantly increased by a high-fat diet. However, ingesting CPPs tended to decrease epididymal fat weight, and retroabdominal fat weight significantly decreased (Fig. 2). 

(Page14, Line 255 to 256)

2) The data of serum glucose concentrations are not a core part of our research, so we changed as following,

 Effect of CPPs on serum glucose and total cholesterol levels of mice

 There were no significant differences in serum glucose concentrations between groups (data not shown). In this experiment, the serum total cholesterol level of the mice, which was significantly increased by the high-fat diet, tended to decrease by feeding the mice CPP (Light) (Fig. 3).

 ↓

 Effect of CPPs on lipid metabolism of mice

 To examine the effect of CPPs on lipid metabolism, we measured the serum total cholesterol levels of the mice. The serum total cholesterol level, which was significantly increased by the high fat diet, tended to decrease by feeding the mice with CPP (Light) (Fig. 3). We also performed H&E staining of the liver and observed that in the HF group, lipid droplets accumulated in hepatocytes, and mice exhibited fatty liver. In all CPP-fed groups, fatty liver was remarkably suppressed (Fig. 4A). These results were consistent with the liver TG levels (Fig. 4B). These results suggest that CPPs inhibit fat accumulation in the liver induced by a high fat diet.

(Page15, Line 266 to 273)

3) We added the new figure as Fig.6.

Effect of CPPs (Light and Extra Dark) treatment on the viability of 3T3-L1 preadipocytes

 Neither type of CPPs (Light or Extra Dark) had any effect on cell viability at concentrations of up to 100 μg/ml (data not shown).

 ↓

Effect of CPPs (Light and Extra Dark) treatment on the viability of 3T3-L1 preadipocytes

 Cell viability of 3T3-L1 preadipocytes was examined with CPPs (Light and Extra Dark) by Neutral red assay. As shown in Fig. 6A and B, neither type of CPP (Light or Extra Dark) had any effect on cell viability at concentrations of up to 100 g/ml. These results showed that CPPs are not cytotoxic to 3T3-L1 preadipocytes.

(Page17, Line 311 to 316)

5. Please include a caption for figures 13A-D. 

6. Please ensure that you refer to Figures 13A-D in your text as, if accepted, production will need this reference to link the reader to the figure.

 → We are grateful for pointing out our carelessness. We removed incorrect figure legends, and changed correct figure legends of Fig. 7-12.

Responses to “Comments”:

 We would like to thank your constructive suggestions.

There is a major problem in lining up your figures/data with the descriptions in the text. For example on line 294, it states that Fig 7 displays the effect of CPP (Light) on GPDH activity. However, in the text figures these enzyme activity results are in Fig 8. The problem appears to be that the data for CPP treatment on TG accumulation was to be Fig 6, but in the MS Fig 6 is about adiponectin and the data are missing. From that point on all figures are off, ie Fig 6 is really Fig 7 etc. These presentation errors must be corrected. Somehow the adiponectin results seem to have been deleted from the MS, but the Figure #s were never adjusted.

 → We are grateful for pointing out our carelessness. We removed incorrect figure legends, and changed correct figure legends of Fig. 7-12. We removed the figure legend of adiponectin results.

Overall the presentation appears somewhat "unconcise", this reviewer did not like a short results description in the text and this was followed by the legend of the Fig for those results in the complete MS.

 → Thank you very much for your valuable suggestions.

 We added the sentences as following, 

Effect of CPPs on body weight 

 To determine the effect of a diet containing CPPs in C57BL/6J mice fed a high fat diet, we measured the body weights of the mice. The weight increased significantly in the HF group compared with the control group. However, there was a tendency for body weight to decrease in the CPP intake group with the HF group. In particular, in the HF + 0.06 L group, there was a significant decrease compared with the HF group (Fig. 1). 

(Page13, Line 241 to 246)

Effect of CPPs on visceral fat weight of mice

 To examine whether the body weight gain in CPP-treated groups resulted in decreased fat accumulation, the visceral fat weight was examined. Visceral fat weight was significantly increased by a high fat diet. However, ingesting CPPs tended to decrease apididymal fat weight, and retroabdominal fat weight significantly decreased (Fig. 2). 

(Page 14, Line 254 to 258)

Effect of CPPs on lipid metabolism of mice

 To examine the effect of CPPs on lipid metabolism, we measured the serum total cholesterol levels of the mice. The serum total cholesterol level, which was significantly increased by the high fat diet, tended to decrease by feeding the mice with CPP (Light) (Fig. 3). We also performed H&E staining of the liver and observed that in the HF group, lipid droplets accumulated in hepatocytes, and mice exhibited fatty liver. In all CPP-fed groups, fatty liver was remarkably suppressed (Fig. 4A). These results were consistent with the liver TG levels (Fig. 4B). These results suggest that CPPs inhibit fat accumulation in the liver induced by a high fat diet. 

(Page 15, Line 266 to 273)

Effect of CPPs (Light and Extra Dark) treatment on the viability of 3T3-L1 preadipocytes

 Cell viability of 3T3-L1 preadipocytes was examined with CPPs (Light and Extra Dark) by Neutral red assay. As shown in Fig. 6A and B, neither type of CPP (Light or Extra Dark) had any effect on cell viability at concentrations of up to 100 g/ml. These results showed that CPPs are not cytotoxic to 3T3-L1 preadipocytes. (Page 17, Line 311 to 321)

Effect of CPP (Light) on GPDH activity in 3T3-L1 preadipocytes

 To ascertain the reduction of TG accumulation in 3T3-L1 preadipocytes treated with CPPs, we examined the effect of CPP (Light) on GPDH activity, a rate-limiting enzyme in TG synthesis. As shown in Fig. 8, GPDH activity was significantly reduced, depending on the concentration of CPP (Light). 

 (Page 18, Line 338 to Page19, Line 342)

About English grammar in this manuscript, we had English editing before submitting by Elsevier language Editing services (Date: 06-Feb-2021, Serial number: LEEX-12443-8FF6610781E7).

Thank you very much for your suggestion, again.

---

## [Decision Letter · Decision Letter 1]

19 Feb 2021

Carob pod polyphenols suppress the differentiation of adipocytes through posttranscriptional regulation of C/EBPß

PONE-D-20-38421R1

Dear Dr. Kojima-Yuasa,

We’re pleased to inform you that your manuscript has been judged scientifically suitable for publication and will be formally accepted for publication once it meets all outstanding technical requirements.

Kind regards,

Juan J Loor

Academic Editor

PLOS ONE

Additional Editor Comments (optional):

Reviewers' comments:

Reviewer's Responses to Questions

**Comments to the Author**

1. If the authors have adequately addressed your comments raised in a previous round of review and you feel that this manuscript is now acceptable for publication, you may indicate that here to bypass the “Comments to the Author” section, enter your conflict of interest statement in the “Confidential to Editor” section, and submit your "Accept" recommendation.

Reviewer #1: All comments have been addressed

2. Is the manuscript technically sound, and do the data support the conclusions?

Reviewer #1: Yes

3. Has the statistical analysis been performed appropriately and rigorously? 

Reviewer #1: Yes

4. Have the authors made all data underlying the findings in their manuscript fully available?

Reviewer #1: Yes

5. Is the manuscript presented in an intelligible fashion and written in standard English?

Reviewer #1: Yes

6. Review Comments to the Author

Reviewer #1: Dear Authors:

All the confusion about the figures etc is now resolved.

On line 257, epididymal is mispelled as apididymal

The Scheme ! is helpful as it indicates how adipogenesis/diff can start but then lipogenesis and fat synthesis can be inhibited long term

7. PLOS authors have the option to publish the peer review history of their article (what does this mean?). If published, this will include your full peer review and any attached files.

Reviewer #1: No

---

## [Editor Report · Acceptance letter]

25 Feb 2021

PONE-D-20-38421R1 

Carob pod polyphenols suppress the differentiation of adipocytes through posttranscriptional regulation of C/EBPß 

Dear Dr. Kojima-Yuasa:

I'm pleased to inform you that your manuscript has been deemed suitable for publication in PLOS ONE. Congratulations! Your manuscript is now with our production department. 

Kind regards, 

on behalf of

Dr. Juan J Loor 

Academic Editor

PLOS ONE